# Anchor–Based Conformal Prediction Under Noisy Annotations in Single–Cell Data

## Abstract

Conformal prediction provides a flexible framework for quantifying prediction uncertainty and has attracted extensive interest. However, most existing methods are designed to handle clean data and may fail to perform satisfactorily when labels are noisy. In this work, we consider the setting where the ground–truth labels are unobserved but crowdsourced noisy labels are available. We introduce an anchor–based conformal prediction method that provides uncertainty quantification. Our method identifies anchor points by selecting samples with strong agreement across annotators. These anchors points are used to train a base predictor that is calibrated to construct a conformal prediction set with a desired coverage rate. Meanwhile, we provide a theoretical analysis of anchor–point identification and provide associated conditions that have been importantly overlooked in the literature. We apply the proposed method to analyze two single–cell datasets to demonstrate its utility and promise.

## 1 Introduction

Conformal prediction emerges as a model–agnostic framework that has attracted extensive attention in supervised learning. It produces prediction sets (for classification tasks) or prediction intervals (for regression problems) with prediction error rate controlled under a desired tolerance level. Conformal prediction may be strategically categorized as full conformal prediction (also referred to as transductive conformal prediction) and split conformal prediction (also called inductive conformal prediction), as discussed in Vovk et al. (2005) and Barber et al. (2023), among others. Various conformal prediction methods have been developed to address different learning objectives. These methods include conformalized quantile regression (Romano et al., 2019), distributional conformal prediction (Chernozhukov et al., 2021), cross–validation+ and jackknife+ (Romano et al., 2020), multi–label outputs (Cauchois et al., 2021), graph neural networks (Zargarbashi et al., 2023), covariate shift (Tibshirani et al., 2019), label shift (Podkopaev & Ramdas, 2021), conformalized survival analysis (Candes et al., 2023), and class–conditional conformal prediction (Ding et al., 2023). For details, see Vovk et al. (1999), Shafer & Vovk (2008), Angelopoulos & Bates (2023), and Fontana et al. (2023).

While those methods provide useful tools to characterize prediction uncertainty, they are typically developed for clean data. They can be vulnerable to perturbations of clean input examples, as examined by Ghosh et al. (2023). On the other hand, in the absence of clean labels, Einbinder et al. (2024) studied the impact of label noise on the validity of conformal prediction, and their analysis suggested that ignoring label noise effects can lead to invalid conformal prediction results. As acquiring accurately annotated data can be expensive or even impossible, it is often of interest to study conform predication for data with noisy or ambiguous labels. For example, concerning image classification, Angelopoulos et al. (2020) described a method for constructing prediction sets from a pre–trained image classifier that are regularized to calibrate unlikely classes. Penso & Goldberger (2024) and Penso et al. (2025) developed conformal prediction methods to handle medical imaging classification networks, where labels are assumed to be corrupted by uniform noise with a known noise level. In many applications, label information is derived from multiple expert annotations, where the majority–voted label is commonly treated as the ground–truth label. When experts seriously disagree, summarizing the expert annotations by a single one–hot distribution can lead to severely deteriorated prediction results. To address this, Stutz et al. (2023) developed Monte Carlo conformal prediction procedures to account for uncertainty associated with ambiguous labels.

Constructing credal regions in a conformal way, Caprio et al. (October 2024) extended classical conformal prediction to problems with ambiguous ground truth, where the exact labels for inputs are not known.

## 1.1 MOTIVATING SETTING

In single–cell transcriptomics, tens of thousands of genes are measured across hundreds of thousands of cells to reveal the information on cell types, subtypes, and states for a tissue sample (Baron et al., 2016). Manual annotations to determine cell types are time–consuming when the number of cells and samples are substantial, and the process can be irreproducible due to varying levels of annotators' expertise. Practically, clustering algorithms are devised to conduct automatic cell identification, e.g., SingleCellNet (Tan & Cahan, 2019) and ACTINN (Ma & Pellegrini, 2020). The automation process, however, involves various challenges, including difficulties in biological interpretation and implementation variability, as discussed by Kiselev et al. (2019). Abdelaal et al. (2019) compared the performance of twenty–two classification methods that automatically assign cell identities for twenty–seven publicly available single–cell RNA sequencing datasets, which differ in sizes, technologies, species, and levels of complexity. While those methods output overlapping classes, their performance varies and typically depends on the data complexity. Different methods often yield varying cluster numbers and cell assignments, and it is important to address uncertainty in identified labels from automatic annotation methods rather than merely take results of a single clustering method as ground truth labels. To this end, we cast the problem into the conformal prediction framework by treating output proxy labels from multiple automatic annotation algorithms as crowdsourced labels (Ibrahim et al., 2023). Utilizing the research on noisy labels, we introduce a new conformal prediction method to handle data with crowdsourced labels.

## 1.2 OUR CONTRIBUTIONS

Without restricting to a specific model for label noise, we consider general cases where label noise can be instance–dependent, and utilize deep neural network architectures to provide a flexible representation of the annotation process to reflect annotator skills as well as possible influence of the true class labels, in addition to the dependence on the input. We utilize the notation of anchor points (Xia et al., 2019), defined in Section 2.1, to bypass the need to access the true labels in order to estimate the instance–dependent noise transition matrix. Although anchor points can be heuristically identified based on applying majority–voting to data with noisy labels (Liu & Tao, 2016; Patrini et al., 2017), this method is only valid under certain conditions, which, however, are unidentified in existing work. In this work, we close this gap and further make the following contributions:

- We provide a necessary–and–sufficient characterization for anchor points, which is accompanied by an identified, mild condition for annotation. We further identify conditions that ensure the validity of using majority–voting to find anchor points from corrupted data. These analyses provides theoretical insights into the available works that utilize anchor points and make them valid for settings satisfying those identified conditions.

- We extend the conformal prediction framework to accommodate corrupted labels and develop true–label prediction sets by using crowdsourced noisy annotations. We introduce an anchor–based method that couples with flexile deep neural models to learn complex annotation processes, which commonly arise from biomedicine and other fields.

- We establish theoretical guarantees and validate the method on two single–cell RNA–seq datasets to demonstrate its utility. Although we focus on single–cell classification, the framework applies broadly to general biomedical applications or problems with crowdsourced noisy labels.

In summary, we integrate conformal prediction and crowdsourced noisy labels to provide valid prediction sets. Our work supplies a new addition to address robustness of conformal prediction to label noise, which enjoys broad applications involving ambiguous or noisy labels.

## 2 PROBLEM SETUP AND METHODS

For $i \in [n]$, let $x_i \in \mathcal{X} \subseteq \mathbb{R}^p$ denote the feature vector for cell $i$ (e.g., gene expression), associated with an unobserved true class $y_i \in [K]$ (e.g., cell type), where $[K] = \{1, \dots, K\}$ is the label space and $\mathcal{X}$ is the input space of the $p$–dimension. For each cell we observe a vector of noisy labels $\widetilde{y}_i = (\widetilde{y}_i^{(1)}, \dots, \widetilde{y}_i^{(R_i)})$ from $R_i$ annotators; for ease of exposition we assume $R_i \equiv R$ (which matches our application datasets). We use uppercase letters $X_i$, $Y_i$ and $\widetilde{Y}_i$ (with the subscript $i$ sometimes omitted) for the corresponding random variables. Let $\mathbb{P}$ represent the probability or conditional probability for the associated random variables. We are interested in devising a conformal prediction method by utilizing the notion of anchor points.

### 2.1 ANCHOR POINTS

**Definition 2.1.** An instance $x \in \mathcal{X}$ is called an anchor point for class $k \in [K]$ if
$$\mathbb{P}(Y = k \mid X = x) = 1.$$

Anchor points were considered by Liu & Tao (2016), Patrini et al. (2017), Xia et al. (2019), and Guo et al. (2023), among others. Its introduction ensures the identifiability of the transition matrix that is formed by the conditional probability of $\widetilde{Y}^{(r)}$ given $Y$ and $X$, which, as shown in the following theorem, can be learned from training data consisting of anchor points and the associated noisy annotated labels alone, provided suitable conditions.

**Assumption 1.** *For any* $r \in [R]$, $k \in [K]$, *and* $x \in \mathcal{X}$,
$$\mathbb{P}(\widetilde{Y}^{(r)} = k | Y = k, X = x) > \mathbb{P}(\widetilde{Y}^{(r)} = k | Y = j, X = x) \quad \textit{for all} \quad j \neq k.$$

This assumption indicates that given the input, the true label is more likely to be annotated than any other labels. In other words, annotators have reasonably competent skills while they may be unable to annotate the true labels surely.

**Theorem 1.** *Suppose Assumption 1 holds for the annotation process. Then for* $x \in \mathcal{X}$,
$$\mathbb{P}(Y = k | X = x) = 1 \quad \textit{if and only if} \quad \mathbb{P}(\widetilde{Y}^{(r)} = k | X = x) = \mathbb{P}(\widetilde{Y}^{(r)} = k | Y = k, X = x).$$

The equivalence in Theorem 1 allows us to use anchor points to learn the transition matrix, even though our training data have no information about the true labels. A natural question then arises: with the availability of corrupted–labeled data only, how to find anchor points? Li et al. (2021) suggested to take $\arg\max_x \mathbb{P}(\widetilde{Y} = k \mid X = x)$ as anchor points for class $k$, provided noisy class–posterior $\mathbb{P}(\widetilde{Y} = k \mid X = x)$ is accurately modeled and the data size is sufficient. This majority–voting scheme echoed the proposals of Liu & Tao (2016) and Patrini et al. (2017), which, however, is not theoretically justified. Its validity is not automatic and requires certain conditions. To close this gap, we identify conditions that allow the use of the majority–voting scheme, which are importantly overlooked in the existing literature.

**Assumption 2.** *Assume that*
$$\mathbb{P}(\widetilde{Y}^{(r)} = k \mid Y = k, X = x(k)) = \mathbb{P}(\widetilde{Y}^{(r)} = k \mid X = x(k)) \tag{1}$$
*holds for* $x(k) = \mathrm{argmax}_x \mathbb{P}(\widetilde{Y}^{(r)} = k \mid X = x)$, *with* $k \in [K]$ *and* $r \in [R]$.

Assumption 2 essentially states that $x(k)$ is sufficient for predicting $\widetilde{Y}^{(r)}$ to be $k$ and captures all relevant information about the true label $Y = k$. This assumption is often plausible in applications. In email spam detection, for example, let $Y$ denote the true label (spam or not spam), $\widetilde{Y}^{(r)}$ be the classifier's prediction, and $X$ include features extracted from emails (e.g.,the number of links, presence of certain words, etc.). If the features are sufficiently informative, then the assumption (1) is feasible. Importantly, (1) is required only for those selected points $x(k)$, but not for all $x \in \mathcal{X}$. This condition is weaker than the global *nondifferential misclassification* condition
$$\mathbb{P}\left(\widetilde{Y}^{(r)} = j \mid Y = k, X = x\right) = \mathbb{P}\left(\widetilde{Y}^{(r)} = j \mid X = x\right) \quad \text{for all } j, k \in [K],$$
which requires conditional independence between $\widetilde{Y}^{(r)}$ and $Y$, given $X$.

**Theorem 2.** *Suppose Assumptions 1– 2 hold. Then* $\mathrm{x}(k)$ *is an anchor point for class* $k$.

Theorem 2 directly comes from Theorem 1. It suggests that applying majority–voting, anchor points can be discovered directly from corrupted data without observing true labels. Combining Theorems 1 and 2 asserts that utilizing anchor points, the instance–dependent transition matrix can be estimated even though we have no access to clean labels, and this is the foundation for the subsequent development.

## 2.2 SPLIT CONFORMAL PREDICTION

This subsection presents the conformal prediction procedure by using anchor points to incorporate the label noise effects. We employ deep neural network architectures to model the transition matrix and implement the likelihood method to learn model parameters. Then we calculate calibration scores to construct prediction sets for test data, as detailed below.

**Warm–up and pipeline.** In the warm–up stage, we apply majority–voting to the data $\{\mathrm{x}_i, \widetilde{\mathrm{y}}_i\}_{i \in [n]}$ to obtain anchor points. Let $\mathcal{A}_k := \{i \in [n] : \mathbb{P}(\mathrm{Y} = k \mid \mathrm{X} = \mathrm{x})\}$ denote the index set of anchor points for class $k$, and write $\mathcal{A} := \cup_{k \in [K]} \mathcal{A}_k$. For each $k \in [K]$, randomly split $\mathcal{A}_k$ into $\mathcal{A}_k^t$ for training and $\mathcal{A}_k^c$ for calibration (i.e., a hold–out set), and form training and calibration data:

$$\mathcal{D}^t := \bigcup_{k \in [K]} \{(\mathrm{x}_i, \widetilde{\mathrm{y}}_i, y_i{=}k) : i \in \mathcal{A}_k^t\}; \ \mathcal{D}^c := \bigcup_{k \in [K]} \{(\mathrm{x}_i, \widetilde{\mathrm{y}}_i, y_i{=}k) : i \in \mathcal{A}_k^c\}.$$

Let $n^t$ and $n^c$ denote the size of $\mathcal{D}^t$ and $\mathcal{D}^c$, respectively.

**Annotator transition model.** We model the annotation process via two functions, $\psi^{\mathrm{A}}(\mathrm{x})$ and $\psi^{\mathrm{c}}(\mathrm{x})$, which are charaterized by feedforward neural networks, and let $\theta_{\mathrm{A}}$ and $\theta_{\mathrm{c}}$ denote the associated parameters. For annotator $r \in [R]$ and class $j \in [K]$, the probability for annotator $r$ to report class $j$, given the true class $k$ and input x, is modeled by

$$\mathbb{P}\left(\widetilde{\mathrm{Y}}^{(r)} = j \mid \mathrm{Y} = k, \mathrm{X} = \mathrm{x}\right) = \frac{\exp\{\langle \alpha_j^{(r)}, \psi^{\mathrm{A}}(\mathrm{x})\rangle + \langle \beta_j^{(k)}, \psi^{\mathrm{c}}(\mathrm{x})\rangle\}}{\sum_{\ell=1}^K \exp\left\{\langle \alpha_\ell^{(r)}, \psi^{\mathrm{A}}(\mathrm{x})\rangle + \langle \beta_\ell^{(k)}, \psi^{\mathrm{c}}(\mathrm{x})\rangle\right\}}, \quad (2)$$

where $\alpha^{(r)} := \{\alpha_j^{(r)}\}_{j=1}^K$ facilitates annotator–specific effects and $\beta^{(k)} := \{\beta_j^{(k)}\}_{j=1}^K$ captures class–specific structures. Let $\theta = \{\theta_{\mathrm{A}}, \theta_{\mathrm{c}}, \{\alpha^{(r)}\}_{r=1}^R, \{\beta^{(k)}\}_{k=1}^K\}$ denote the resulting model parameters, and let $P_\theta\left(\widetilde{\mathrm{Y}}^{(r)} = j \mid \mathrm{Y} = k, \mathrm{X} = \mathrm{x}\right)$ denote the probability (2) parameterized by $\theta$.

**Anchors–based likelihood training.** Using $\mathcal{D}^t$, we estimate the model parameter $\theta$ by maximizing the log–likelihood function:

$$\widehat{\theta} = \arg\max_\theta \sum_{k \in [K]} \sum_{i \in \mathcal{A}_k^t} \sum_{r=1}^R \sum_{j=1}^K \mathbb{1}\{\widetilde{\mathrm{y}}_i^{(r)} = j\}\log P_\theta\left(\widetilde{\mathrm{Y}}^{(r)} = j \big| \mathrm{Y} = k, \mathrm{X} = \mathrm{x}_i\right), \quad (3)$$

where $\mathbb{1}(\cdot)$ is the indicator function. Using the estimated parameter $\widehat{\theta}$, for $r \in [R]$ and $k, j \in [K]$, we define $\widehat{\tau}_{kj}^{(r)}(\mathrm{x}) := P_{\widehat{\theta}}\left(\widetilde{\mathrm{Y}}^{(r)} = j \big| \mathrm{Y} = k, \mathrm{X} = \mathrm{x}\right)$ and

$$\widehat{\tau}_k(\mathrm{x}_i, \widetilde{\mathrm{y}}_i) := \prod_{r=1}^R \prod_{j=1}^K \left\{\widehat{\tau}_{kj}^{(r)}(\mathrm{x}_i)\right\}^{\mathbb{1}\{\widetilde{\mathrm{y}}_i^{(r)}=j\}}. \quad (4)$$

**Conformal Prediction.** To construct a prediction set for a test data with a $(1-\alpha)$ coverage rate for $\alpha \in (0, 1)$, we first determine the threshold value using the held-out data $\mathcal{D}^c$ by proceeding with the following five steps: (i) for each $i \in \mathcal{A}^c := \bigcup_{k \in [K]} \mathcal{A}_k^c$, compute $\{\widehat{\tau}_k(\mathrm{x}_i, \widetilde{\mathrm{y}}_i)\}_{k \in [K]}$; (ii) sort them in decreasing order (assuming no ties) and let $\widehat{y}_{(1)}(\mathrm{x}_i, \widetilde{\mathrm{y}}_i), \cdots, \widehat{y}_{(K)}(\mathrm{x}_i, \widetilde{\mathrm{y}}_i)$ denote the corresponding class labels; (iii) form nested sets $\mathcal{C}(\mathrm{x}_i, \widetilde{\mathrm{y}}_i; k) := \{\widehat{y}_{(1)}(\mathrm{x}_i, \widetilde{\mathrm{y}}_i), \cdots, \widehat{y}_{(k)}(\mathrm{x}_i, \widetilde{\mathrm{y}}_i)\}$ for all $k \in [K]$; (iv) define the calibration score $S(\mathrm{x}_i, \widetilde{\mathrm{y}}_i; \mathrm{y}_i) = \min\{k \in [K] : \mathrm{y}_i \in \mathcal{C}(\mathrm{x}_i, \widetilde{\mathrm{y}}_i; k)\}$ for all $i \in \mathcal{A}^c$; and (v) define $\widehat{k}^c(\alpha) = \min\{k \in [K] : |\{i \in \mathcal{A}^c : s(\mathrm{x}_i, \widetilde{\mathrm{y}}_i; \mathrm{y}_i) \leq k\}| \geq (n^c + 1)(1 - \alpha)\}$.

Next, for prediction of a test data point with $(x, \widetilde{y})$, we repeat Steps (i)–(iii) with $(x_i, \widetilde{y}_i)$ replaced by $(x, \widetilde{y})$. The predicted label $\widehat{y}$ for the test data corresponds to the class having the largest value derived from applying (4) to $(x, \widetilde{y})$, and the prediction set is taken as

$$\mathcal{C}_\alpha(x, \widetilde{y}) := C(x, \widetilde{y}; \widehat{k}^c(\alpha)),$$

by setting $k$ in Step (iii) to $\widehat{k}^c(\alpha)$. We call the resulting set a top-$k$ conformal set and summarize this procedure in Algorithm 1.

Our procedure is developed for settings where for given features, annotators label instances independently, i.e., the following assumption is made:

**Assumption 3.** $\mathbb{P}(\widetilde{Y} = \widetilde{y} \mid X = x) = \prod_{r=1}^{R} \mathbb{P}(\widetilde{Y}^{(r)} = \widetilde{y}^{(r)} \mid X = x)$.

This assumption is not essential yet it simplifies learning of the joint label–noise process for all annotators to estimating the annotator–specific transition model (2). When this assumption is deemed infeasible, one may modify the procedure by learning the joint model $\mathbb{P}(\widetilde{Y} = \widetilde{y} \mid X = x)$ or incorporate a shared latent factor to capture dependence across annotators. It is worth noting that variables $\{X_i, \widetilde{Y}_i, Y_i\}_{i \in \mathcal{A}^c}$ do need not be identically distributed. Further, as the top-$k$ method takes anchor points as input, one might wonder how variability in identifying anchor points, together with uncertainty in model specification and the estimation of model parameters using the likelihood method, may affect statistical guarantees of the resulting prediction sets. This is also a natural concern arising from existing conformal prediction methods, which typically involves multiple stages of determining intermediate quantities. Fortunately, as noted by Romano et al. (2020), these sources of variability are automatically accounted for through the threshold $\widehat{k}^c(\alpha)$, which is chosen adaptively to ensure finite-sample coverage on future test points, as shown in the proof of Theorem 3 in Appendix A.2.

---

**Algorithm 1:** Anchor–Based Conformal Prediction

**Input:**

        Anchor-based training and calibration data: $\mathcal{D}^t$ and $\mathcal{D}^c$

        Target miscoverage rate $\alpha \in (0, 1)$; Test data $(x, \widetilde{y})$

**Training based on $\mathcal{D}^t$:**

        Solve (3) and obtain $\widehat{\theta}$

**Determine Threshold Value using $\mathcal{D}^c$:**

    1. For all $i \in \mathcal{A}^c$,

       (i) For all $k \in [K]$, compute $\widehat{\tau}_k(x_i, \widetilde{y}_i)$ by (4);

      (ii) Sort the values in decreasing order, and let $\widehat{y}_{(1)}(x_i, \widetilde{y}_i), \cdots, \widehat{y}_{(K)}(x_i, \widetilde{y}_i)$ denote the corresponding class labels

      (iii) For all $k \in [K]$, form nested sets $\mathcal{C}(x_i, \widetilde{y}_i; k) = \{\widehat{y}_{(1)}(x_i, \widetilde{y}_i), \cdots, \widehat{y}_{(k)}(x_i, \widetilde{y}_i)\}$

      (iv) Define the calibration score $S(x_i, \widetilde{y}_i; k) = \min\{k \in [K] : y_i \in \mathcal{C}(x_i, \widetilde{y}_i; k)\}$

    2. Set $\widehat{k}^c(\alpha) = \min\{k \in [K] : |\{i \in \mathcal{A}^c : S(x_i, \widetilde{y}_i; y_i) \le k\}| \ge (n^c + 1)(1 - \alpha)\}$

**Output:** Prediction Set for Test Data

        $C_\alpha(x, \widetilde{y}) = \mathcal{C}(x, \widetilde{y}; \widehat{k}^c(\alpha))$

---

## 3   Theoretical Guarantee

For brevity, we write $p_x(y) := \mathbb{P}(Y = y \mid X = x)$, $q_x^{(r)}(j \mid y) := \mathbb{P}(\widetilde{Y}^{(r)} = j \mid Y = y, X = x)$, and $P_x(y, \widetilde{y}) := \mathbb{P}(Y = y, \widetilde{Y} = \widetilde{y} \mid X = x)$. We examine theoretical results for the proposed conformal prediction method, provided certain assumptions.

**Assumption 4.** *Calibration scores $\{S(X_i, \widetilde{Y}_i; Y_i)\}_{i \in \mathcal{A}^c}$ for the calibration anchor points and $S(X, \widetilde{Y}; Y)$ for a test point are exchangeable and almost surely distinct, or ties are broken at random.*

**Assumption 5.** *For* $y, j \in [K]$ *and* $r \in [R]$, $p_x(y)$ *and* $q_x^{(r)}(j \mid y)$ *are L-Lipschitz in* x *with respect to a given norm* $\| \cdot \|$.

**Theorem 3.** *Suppose that Assumptions 1– 4 hold and that* $\widehat{\theta}$ *in (3) is a consistent estimator of* $\theta$. *Then for any* $\alpha \in (0, 1)$,

$$1 - \alpha \leq \mathbb{P}\{Y \in C_\alpha(X, \widetilde{Y})\} < 1 - \alpha + \frac{1}{n^c + 1}.$$

Theorem 3 establishes the marginal coverage rate for the proposed prediction set, which is bounded between $1 - \alpha$ and $1 - \alpha + \frac{1}{n^c+1}$. If the calibration anchor point set is sufficiently large, the coverage rate is almost identical to $1 - \alpha$. Further, one may be interested in evaluating the conditional coverage rate of the prediction set $C_\alpha(x, \widetilde{y})$, a stronger version than the marginal coverage: Is it true that $P\{Y \in C_\alpha(X, \widetilde{Y}) \mid X = x\} \geq 1 - \alpha$ for $x \in \mathcal{X}$? This question is about valid coverage conditional on a specific observed value of the feature $X$. However, as noted by Lei & Wasserman (2014), Vovk (2012), and Barber et al. (2021), finite sample conditional validity is impossible for any distribution $\mathbb{P}$ and any $x \in \mathcal{X}$ unless x is an atom, as defined in Lei & Wasserman (2014). That said, it is undeniable that conditional coverage would be preferable. We thus take a step back to relax the requirement of conditional validity by considering a weaker condition and introduce the following definition, which can be regarded as an approximate conditional coverage.

**Definition 3.1.** For $\alpha \in (0, 1)$ and $\gamma \in [0, 1)$, an instance x is called an $(\alpha, \gamma)$-*conditional valid point* with respect to $C_\alpha(x, \widetilde{y})$ if $\mathbb{P}\{Y \in C_\alpha(X, \widetilde{Y}) \mid X = x\} \geq 1 - \alpha - \gamma$. Let the collection of those $(\alpha, \gamma)$-*conditional valid points* be denoted

$$\mathcal{V}_{\alpha, \gamma} := \{x \in \mathcal{X} : \mathbb{P}\{Y \in C_\alpha(X, \widetilde{Y}) \mid X = x\} \geq 1 - \alpha - \gamma\}.$$

For $\rho > 0$, define $\mathcal{N}_{\alpha, \gamma}(\rho) = \left\{x \in \mathcal{X} : \inf_{x_0 \in \mathcal{V}_{\alpha, \gamma}} \|x - x_0\| \leq \rho\right\}$ to be the $\rho$-neighborhood of $\mathcal{V}_{\alpha, \gamma}$.

**Proposition 1.** Suppose the assumptions in Theorem 3 and Assumption 5 hold. Then for $\alpha \in (0, 1)$ and $\gamma \in [0, 1)$,

$$\mathbb{P}(X \in \mathcal{V}_{\alpha, \gamma}) \geq 1 - \frac{\alpha}{\alpha + \gamma}.$$

**Theorem 4.** *Suppose the assumptions in Theorem 3 and Assumption 5 hold, and let* $\alpha, \delta \in (0, 1)$ *and* $\rho \in [0, 1)$. *Then for any* $x \in \mathcal{N}_{\alpha, \gamma}(\rho)$,

$$\mathbb{P}\{Y \in C_\alpha(X, \widetilde{Y}) \mid X = x\} \geq 1 - \alpha - \gamma - \frac{1}{2}(1 + R)K^{R+1}L\rho. \tag{5}$$

*Consequently,*

(a). $\mathbb{P}\{Y \in C_\alpha(X, \widetilde{Y}) \mid X \in \mathcal{N}_{\alpha, \gamma}(\rho)\} \geq 1 - \alpha - \gamma - \frac{1}{2}(1 + R)K^{R+1}L\rho$;

(b). $\mathbb{P}\{Y \in C_\alpha(X, \widetilde{Y}) \mid X \in \mathcal{V}_{\alpha, \gamma}\} \geq 1 - \alpha - \gamma$.

Proposition 1 shows that $\mathcal{V}_{\alpha, \gamma}$ has strictly positive probability mass for every $\gamma > 0$, and thus it is nonempty. In particular, $\mathbb{P}(X \in \mathcal{V}_{\alpha, 1-\alpha}) \geq 1 - \alpha$ and $\mathbb{P}(X \in \mathcal{V}_{\alpha, 1/2-\alpha}) \geq 1 - 2\alpha$ if $0 < \alpha < 1/2$, which illustrates that a larger $\gamma$ value tolerates a greater deviation from the conditional coverage level. When $\gamma = 0$, then $\mathcal{V}_{\alpha, 0}$ includes all x values that ensures the conditional validity (if $\mathcal{V}_{\alpha, 0}$ is nonempty). In this case, Theorem 4 describes a weaker conditional coverage for those points not in $\mathcal{V}_{\alpha, \gamma}$ but in its neighborhood. Interestingly, the number of annotators and the number of classes come into play in the lower bound in (5).

## 4 IMPLEMENTATION PROCEDURES

We develop a procedure for predicting latent true labels from multiple noisy annotators. The approach proceeds by constructing per–class anchor sets from high–agreement subsets of annotations, learning an instance–dependent transition model parameterized by two deep networks, forming likelihood–based class scores for point prediction, and calibrating top-$k$ prediction sets on held–out anchors. As illustrated in Figure 1, our method consists of four main stages: anchor selection, base predictor training, anchor–guided calibration, and conformal prediction set generation.

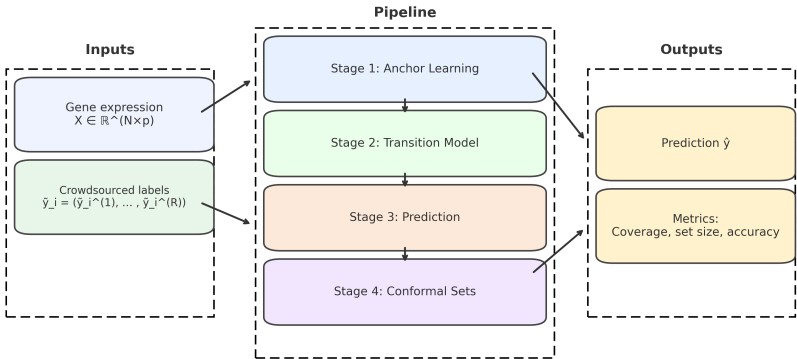

Figure 1: Overview of the proposed anchor-based conformal prediction pipeline. The framework identifies anchors from annotator agreement, trains a base model on these anchors, and calibrates conformal prediction sets to handle label noise.

**Data and Preprocessing.** We evaluated the performance of the proposed top-$k$ method by analyzing two single–cell RNA–seq datasets: *Baron3* (Baron et al., 2016) and *PBMC2* (Stuart et al., 2020). The *Baron3* data were collected from pancreatic islets from human donors. The *PBMC2* data comprise peripheral blood mononuclear cells from healthy donors. Both datasets contain substantial cellular and genetic information and were labeled for classification. In terms of gene counts, they are comparable, with counts ranging from 20,000 to 24,000. The information of gene and annotated cell types was displayed as a matrix, where 1,200 highly variable genes are selected. Counts were transformed using the centered log–ratio to mitigate compositional effects and used to construct a nearest–neighbor graph. Across experiments, we used stratified splits, Adam optimization with early stopping on validation loss, and cross–entropy as the primary objective.

**Base Predictors.** We applied our anchor–based conformal prediction method with several choices of the base predictor for cell–type classification. Within the family of graph neural networks, the graph convolutional network updates node representations by aggregating neighborhood features through learned filters (Gao et al., 2023). The graph attention network extends this by assigning attention weights, allowing the predictor to emphasize more informative neighbors (Liu & Zhou, 2020). GraphSAGE provides an inductive variant that samples neighborhoods and aggregates features to construct low–dimensional node embeddings suitable for large graphs (Hamilton et al., 2018). As a non-graph baseline, we also considered a Multi-Layer Perceptron, a standard feed–forward predictor with fully connected layers (Gharehbaghi, 2023).

## 5   ANALYSIS RESULTS

We employed the proposed anchor–based method to analyze the two datasets, and compared the performance to a baseline of the adaptive prediction sets (APS) method (Romano et al., 2020), a widely used method in practice.

**Baron3.** Agreement–based anchor selection yielded 68 high–confidence cells across 15 cell types. Anchor representation was diverse relatively enriched in T cells (7.4%) and macrophages (6.0%), but sparse in acinar and $\alpha$ cells (both $< 1\%$), reflecting variation in annotator agreement (Table 1a). Using the anchor–calibration split, Top-$k$ conformal sets closely tracked nominal coverage $(1-\alpha)$ at 80%, 85%, 90%, and 95% targets, with empirical coverage results at 0.80, 0.88, 0.95, and 0.95, respectively. By contrast, APS usually achieved highly conservative coverage ($\geq 96\%$ across targets) but produced much larger prediction sets. Visualization on a low–dimensional embedding further supports these patterns: under APS, nearly all cells attain maximal set sizes, whereas our Top-$k$ set sizes vary smoothly across clusters, with smaller sets in well–separated endocrine populations and larger sets in ambiguous ductal and stellate regions (Figure 2).

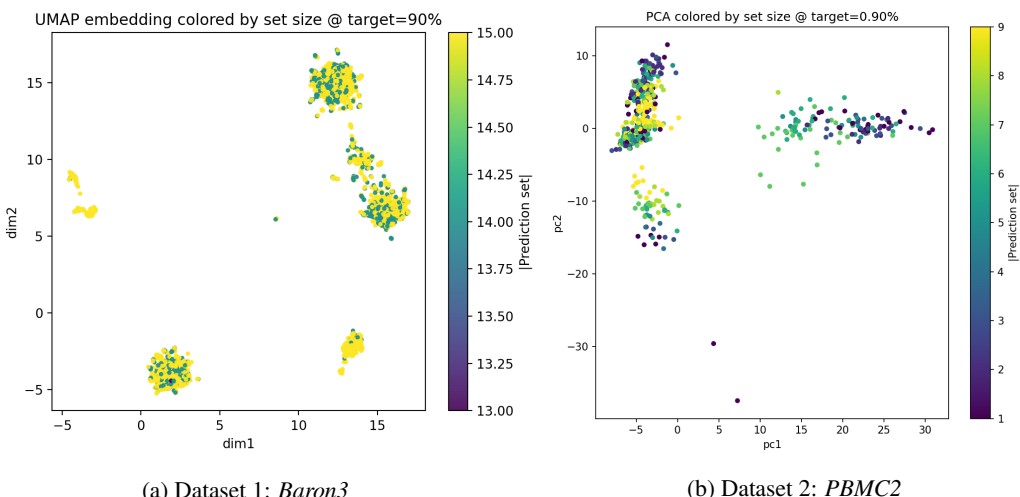

(a) Dataset 1: *Baron3*         (b) Dataset 2: *PBMC2*

Figure 2: Low–dimensional embedding (UMAP or PCA fallback) colored by conformal set size at 90% target. Regions with higher ambiguity receive larger sets; well–separated clusters receive smaller sets.

**PBMC2.** Anchor selection again revealed substantial heterogeneity (Table 1b). Some immune subtypes exhibited relatively high anchor proportions, whereas others had very few, underscoring differences in annotator consistency. As in *Baron3*, APS achieved near–perfect coverage across all nominal levels but at the cost of inflated set sizes, often approaching the entire label space. In contrast, our Top-$k$ delivered coverage much closer to the target values while maintaining smaller, more interpretable sets. Compact sets concentrated in well–defined clusters, whereas ambiguous regions yielded larger sets, as expected. To quantify this trade–off across datasets, Table 2 reports average conformal set sizes. APS consistently produced very large sets (near the total number of classes), whereas Top-$k$ yielded compact and interpretable sets (average sizes 12–14 in *Baron3* and substantially smaller in *PBMC2*). Together with the anchor statistics, these results indicate that anchors not only capture annotator agreement but also enable calibration procedures that produce valid, biologically meaningful, and compact conformal prediction sets. Overall, anchor–guided Top-$k$ method maintains reliable calibration with practical utility, while APS serves as a conservative upper baseline. Additional per–class results, including confusion matrices (Figure 4) and detailed classification metrics (Tables 3–4), are provided in Appendix D.

**Set size analysis across datasets.** Table 2 summarizes the average conformal prediction set sizes obtained from the anchor–based calibration procedure and APS for both datasets. In *Baron3*, APS produced nearly maximal set sizes (14–15 labels on average), confirming its conservative nature and limited informativeness. By contrast, in *PBMC2*, the anchor–based procedure yielded much more compact sets, often of size one across most targets, with only a modest increase at the 95% coverage level. This difference illustrates how anchor prevalence and annotator agreement directly influence calibration outcomes. When anchors are sparse (*Baron3*), conformal sets inflate toward the full label space, whereas when anchors are abundant and reliable (*PBMC2*), prediction sets remain small and interpretable. Together, these results reinforce that anchors provide a flexible mechanism for trading off between validity and efficiency across datasets of differing annotation quality. APS provides conservative coverage with inflated sets, while Top-$k$ achieves coverage close to nominal with more compact and interpretable sets.

Furthermore, we compared the performance of the proposed method to three additional baselines: regularized adaptive prediction sets (RAPS), sorted adaptive prediction sets (SAPS), and split conformal prediction using softmax scores (SoftCP). RAPS (Angelopoulos et al., 2020) and SAPS (Huang et al., 2024) are derived from APS, with the goal to output stable predictive sets. RAPS regularizes conformity scores by adding a penalty function to exclude unlikely classes, whereas SAPS examines softmax probabilities and retains only the maximum probability. SoftCP is a split conformal prediction method (Lei et al., 2018) applying softmax class probabilities and the score–based

Table 1: Anchor counts and proportions (Prop) per cell type for both datasets.

(a) Dataset 1: *Baron3*

| Cell type | Total | Anchors | Prop |
|---|---|---|---|
| t_cell | 81 | 6 | 0.0741 |
| macrophage | 117 | 7 | 0.0598 |
| epsilon | 132 | 7 | 0.0530 |
| mast | 78 | 4 | 0.0513 |
| schwann | 86 | 4 | 0.0465 |
| quiescent_stellate | 96 | 4 | 0.0417 |
| name | 107 | 4 | 0.0374 |
| gamma | 133 | 4 | 0.0301 |
| ductal | 218 | 6 | 0.0275 |
| beta | 334 | 5 | 0.0150 |
| delta | 292 | 4 | 0.0137 |
| endothelial | 160 | 2 | 0.0125 |
| activated_stellate | 565 | 4 | 0.0071 |
| alpha | 457 | 3 | 0.0066 |
| acinar | 737 | 4 | 0.0054 |

(b) Dataset 2: *PBMC2*

| Cell type | Total | Anchors | Prop |
|---|---|---|---|
| B cell | 250 | 215 | 0.8600 |
| cMono | 409 | 349 | 0.8533 |
| ncMono | 119 | 92 | 0.7731 |
| CD4 T cell | 1238 | 872 | 0.7044 |
| NK cell | 270 | 185 | 0.6852 |
| CD8 T cell | 676 | 434 | 0.6420 |
| cDC | 20 | 12 | 0.6000 |
| pDC | 12 | 7 | 0.5833 |
| Plasma cell | 6 | 0 | 0.0000 |

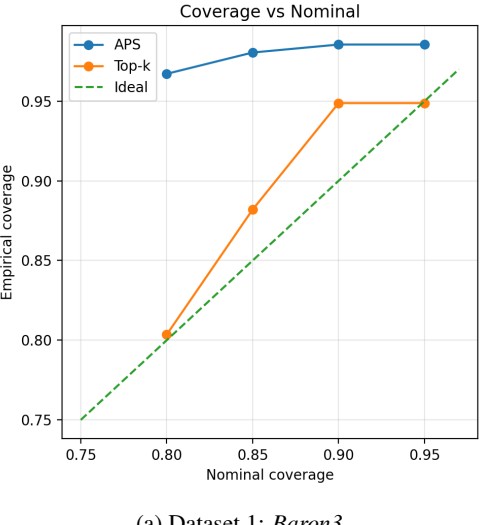

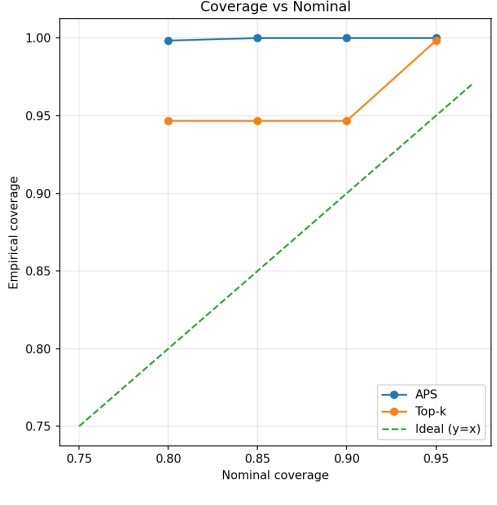

(a) Dataset 1: *Baron3*

(b) Dataset 2: *PBMC2*

Figure 3: Empirical versus nominal coverage for two datasets. APS is highly conservative, while Top-$k$ tracks nominal levels more closely.

conformal classification formulation. We report in Appendix D (Table 6) the analysis results of the empirical coverage rates and the size of the resulting prediction set corresponding to four values of the target size: $1 - \alpha = 0.80, 0.85, 0.90$ and $0.95$. Clearly, no method outperforms others simultaneously with respect to the coverage rate and set size, which aligns with the expectation that as a higher coverage tends to require a larger prediction set. Furthermore, no method exhibits consistently better performance on both datasets, and this underscores that heterogeneity in data plays an important role in affecting the performance of the methods: the performance of a method relies on whether the associated conditions are true or nearly true. For the Baron3 dataset, SoftCP and RAPS yield under–coveraged prediction sets. While the coverage rates produced by SAPS agree with the target rates, they are a lot smaller than those from APS and our top-$k$ method although the latter two methods output slightly larger prediction sets. Regarding the PBMC2 dataset, all methods produce higher coverage rates than the target levels, the size of the resulting prediction sets varies, which can be roughly grouped into two categories: large or small. APS and RAPS produce a lot larger prediction sets, suggesting reduced efficiency. On the other hand, our top-$k$ method, together with

Table 2: Empirical coverage and average set size for APS and Top-$k$ conformal prediction across targets on both datasets.

| Method | Dataset | Target | Empirical Coverage | Avg. Set Size |
|--------|---------|--------|--------------------|---------------|
| APS | *Baron3* | 0.80 | 0.967 | 14.49 |
|  |  | 0.85 | 0.981 | 14.68 |
|  |  | 0.90 | 0.986 | 14.74 |
|  |  | 0.95 | 0.986 | 14.74 |
|  | *PBMC2* | 0.80 | 0.998 | 4.40 |
|  |  | 0.85 | 1.000 | 4.94 |
|  |  | 0.90 | 1.000 | 5.45 |
|  |  | 0.95 | 1.000 | 6.20 |
| Top-$k$ | *Baron3* | 0.80 | 0.804 | 12.00 |
|  |  | 0.85 | 0.882 | 13.00 |
|  |  | 0.90 | 0.949 | 14.00 |
|  |  | 0.95 | 0.949 | 14.00 |
|  | *PBMC2* | 0.80 | 0.947 | 1.00 |
|  |  | 0.85 | 0.947 | 1.00 |
|  |  | 0.90 | 0.947 | 1.00 |
|  |  | 0.95 | 0.998 | 2.00 |

SoftCP and SAPS, gives effective results with the smallest prediction sets. Furthermore, our method yields the highest coverage rates among these three methods, which demonstrates the promise of our method.

## 6 CONCLUSION AND DISCUSSION

We introduced an anchor–based conformal prediction method for classification with crowdsourced noisy annotations. By leveraging anchor points to guide calibration, our method provides rigorous uncertainty quantification. In applications, when the number of anchor points is small or even zero for some classes, we may enlarge $\mathcal{A}$ by including pseudo-anchors: an instance x is called a $\delta$-pseudo-anchor for class $k$ if $\mathbb{P}(Y = k \mid X = x) \geq 1 - \delta$ for $0 \leq \delta < 1$. When $\delta = 0$, it becomes an anchor point; $\delta$-pseudo-anchor points are also called anchor points by Xia et al. (2019) if $\delta$ is close to zero; additional discussions are deferred to Appendix B. A future work is waranted to examine the impact on coverage rates and sizes of conformal prediction sets when anchor points are mis–identified in settings violating the assumptions in Theorems 1 and 2. Examining the exact influence of the number of anchor points can be valuable, although it is expected that, in principle, the more anchor points, the better learning results.

It is worthwhile to further assess the performance of the proposed top-$k$ method from other perspectives. For example, it is interesting to explore how the inclusion of pseudo-anchor points or how different degrees of class imbalance may affect the coverage rate of prediction sets. As observed in Section 5, different methods may perform differently when applied to different data, and this reflects the fact that heterogeneity in data plays an important role in affecting the performance of a method while the application to two single–cell datasets confirmed the promise of our method, it is useful in assessing how the proposed method performs when applied to other settings such as imaging or language processing data.

## 7 ETHICS STATEMENT

This paper introduces an anchor–based conformal prediction approach for handling noisy annotations in single–cell data. The goal of this work is to improve the reliability and robustness of predictive modeling in biomedical research, thereby enhancing our understanding of cellular heterogeneity and disease mechanisms. We have carefully considered the ethical implications and do not anticipate any direct negative consequences arising from this work. While potential downstream applications may involve clinical or biomedical decision–making, this study is methodological in nature and not directly applied to patient care. We are committed to the responsible communication and use of our methods and encourage their application in ways that respect ethical standards in biomedical research and data privacy.

## 8 REPRODUCIBILITY STATEMENT

We have taken steps to ensure the reproducibility of our work. All datasets used in this study are clearly referenced. Descriptions of preprocessing procedures, model architectures, training protocols, and evaluation metrics are provided in the Methods section and Appendix.

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

APPENDICES: TECHNICAL DETAILS AND EXTENDED ANALYSIS RESULTS

# A   PROOFS OF THEOREMS

## A.1   PROOF OF THEOREM 1

First, we comment that in defining anchor points, it is implicitly assumed that an instance x can be an anchor point for at most one class. That is, if $\mathbb{P}(Y = k|X = x) = 1$, then $\mathbb{P}(Y = j|X = x) = 0$ for any $j \neq k$. However, each class can have multiple anchor points; having $\mathbb{P}(Y = k|X = x) = 1$ does not exclude $\mathbb{P}(Y = k|X = x^*) = 1$ for those instances $x^*$ that are not identical to x.

*Proof of Theorem 1:* First we show the '$\Longrightarrow$" direction, which is immediate from the following derivations:

$$
\begin{aligned}
& \mathbb{P}(\widetilde{Y}^{(r)} = \tilde{y}^{(r)}|X = x) \\
= {} & \sum_{j \in \mathcal{Y}} \left\{ \mathbb{P}(\widetilde{Y}^{(r)} = \tilde{y}^{(r)}|Y = j, X = x)\mathbb{P}(Y = j|X = x) \right\} \\
= {} & \sum_{j \neq k} \left\{ \mathbb{P}(\widetilde{Y}^{(r)} = \tilde{y}^{(r)}|Y = j, X = x)\mathbb{P}(Y = j|X = x) \right\} \\
& + \mathbb{P}(\widetilde{Y}^{(r)} = \tilde{y}^{(r)}|Y = k, X = x)\mathbb{P}(Y = k|X = x) \\
= {} & \mathbb{P}(\widetilde{Y}^{(r)} = \tilde{y}^{(r)}|Y = k, X = x), \quad\quad\quad\quad\quad\quad (A.1)
\end{aligned}
$$

where we use the conditions for anchor points.

Next, we show the "$\Longleftarrow$" direction. Indeed, applying (A.1) to the condition

$$ \mathbb{P}(\widetilde{Y}^{(r)} = k|X = x) = \mathbb{P}(\widetilde{Y}^{(r)} = k|Y = k, X = x) $$

leads to

$$
\begin{aligned}
& \sum_{j \neq k} \mathbb{P}(\tilde{Y}^{(r)} = k|Y = j, X = x)\mathbb{P}(Y = j|X = x) \\
& + \mathbb{P}(\widetilde{Y}^{(r)} = k|Y = k, X = x)\{\mathbb{P}(Y = k|X = x) - 1\} = 0,
\end{aligned}
$$

which is equivalently written as

$$ \sum_{j \neq k} \left\{ \mathbb{P}(\tilde{Y}^{(r)} = k|Y = j, X = x) - \mathbb{P}(\widetilde{Y}^{(r)} = k|Y = k, X = x) \right\} \mathbb{P}(Y = j|X = x) = 0. $$

By Assumption 1 and the fact that $\mathbb{P}(Y = j|X = x) \geq 0$ for all $j \neq k$, we conclude that

$$ \mathbb{P}(Y = j|X = x) = 0 \quad \text{for all} \quad j \neq k, $$

and thus yielding

$$ \mathbb{P}(Y = k|X = x) = 1. $$

$\square$

## A.2 PROOF OF THEOREM 3

To prove Theorem 3, we first show the following lemma.

**Lemma 1.** *Suppose $\{c_1, \ldots, c_m; c_{m+1}\}$ is a sequence of constants, taking values in $[K]$. For $k \in [K]$ and $\alpha \in (0, 1)$, define*

$$
\begin{aligned}
\mathcal{I}_k &= \{i \in [m] : c_i \leq k\}, \\
k(\alpha) &= \inf\{k \in [K] : |\mathcal{I}_k| \geq (m+1)(1-\alpha)\}, \text{ and} \\
\mathcal{J} &= \{i \in [m] : c_i < c_{m+1}\}.
\end{aligned}
$$

*Then "$c_{m+1} > k(\alpha)$"    iff    "$|\mathcal{J}| > (m+1)(1-\alpha)$".*

*Proof.* Show "$\Longrightarrow$": For any $i_0 \in \mathcal{I}_{k(\alpha)}$, we have that $c_{i_0} \leq k(\alpha)$. Then by the condition $c_{m+1} > k(\alpha)$, $c_{i_0} < c_{m+1}$, leading to $i_0 \in \mathcal{J}$ by definition of $\mathcal{J}$. Therefore,

$$
\mathcal{I}_{k(\alpha)} \subset \mathcal{J},
$$

yielding $|\mathcal{I}_{k(\alpha)}| \leq |\mathcal{J}|$. Then applying definition of $k(\alpha)$ shows $(m+1)(1-\alpha) \leq |\mathcal{J}|$.

Show "$\Longleftarrow$": We show the conclusion by contradiction. If the conclusion does not hold, then $c_{m+1} \leq k(\alpha)$, implying that $c_i < k(\alpha)$ for any $i \in \mathcal{J}$. Consequently, $\max\{c_i : i \in \mathcal{J}\} < k(\alpha)$. Thus, there exists $k_0$ such that $\max\{c_i : i \in \mathcal{J}\} < k_0 < k(\alpha)$, showing that

$$
\mathcal{J} \subset \mathcal{I}_{k_0}. \tag{A.2}
$$

By the condition $|\mathcal{J}| > (m+1)(1-\alpha)$, we obtain $|\mathcal{I}_{k_0}| > (m+1)(1-\alpha)$. On the other hand, by the definition of $k(\alpha)$, we conclude that $k(\alpha) \leq k_0$, which contradicts (A.2). $\square$

*Proof of Theorem 3.* By definition, for any $(\mathrm{x}, \mathrm{y}) \in \mathcal{D}^c$ and any $k \in [K]$,

$$
S(\mathrm{x}, \widetilde{\mathrm{y}}; \mathrm{y}) \leq k \Longleftrightarrow \mathrm{y} \in \mathcal{C}(\mathrm{x}, \widetilde{\mathrm{y}}; k). \tag{A.3}
$$

Then for any $\alpha \in (0, 1)$,

$$
\begin{aligned}
\widehat{k}^c(\alpha) &= \min\left\{k \in [K] : |\{i \in \mathcal{A}^c : \mathrm{y}_i \in \mathcal{C}(\mathrm{x}_i, \widetilde{\mathrm{y}}_i; k)\}| \geq (n^c + 1)(1-\alpha)\right\} \\
&= \min\left\{k \in [K] : |\{i \in \mathcal{A}^c : S(\mathrm{x}_i, \widetilde{\mathrm{y}}_i; \mathrm{y}_i) \leq k\}| \geq (n^c + 1)(1-\alpha)\right\}
\end{aligned}
$$

Because $\{S(\mathrm{X}_i, \widetilde{\mathrm{Y}}_i; \mathrm{Y}_i)\}_{i \in \mathcal{A}^c}$ and $S(\mathrm{X}, \widetilde{\mathrm{Y}}; \mathrm{Y})$ are exchangeable random variables as stated in Assumption 4, so

$$
U := |\{i \in \mathcal{A}^c : S(\mathrm{X}, \widetilde{\mathrm{Y}}; \mathrm{Y}) > S(\mathrm{x}_i, \widetilde{\mathrm{y}}_i; \mathrm{y}_i)\}|
$$

is stochastically dominated by the discrete uniform distribution on $\{0, 1, \ldots, n^c\}$. When the calibration scores are almost surely distinct (or when random tie-breaking is used to break ties), exchangeability implies that $U$ follows a uniform distribution on $\{0, 1, \ldots, n^c\}$.

Consequently, by (A.3) and Lemma 1,

$$
\begin{aligned}
&\mathbb{P}\{\mathrm{Y} \notin \mathcal{C}(\mathrm{X}, \widetilde{\mathrm{Y}}; k^c(\alpha)\} \\
&= \mathbb{P}\{S(\mathrm{X}, \widetilde{\mathrm{Y}}; \mathrm{Y}) > k^c(\alpha)\} \\
&= \mathbb{P}\{|\{i \in \mathcal{A}^c : S(\mathrm{X}, \widetilde{\mathrm{Y}}; \mathrm{Y}) > S(\mathrm{x}_i, \widetilde{\mathrm{y}}_i; \mathrm{y}_i)\}| > (n^c + 1)(1-\alpha)\} \\
&= \mathbb{P}\{U > (n^c + 1)(1-\alpha)\} \\
&= \sum_{u > (n^c+1)(1-\alpha)} \frac{1}{n^c + 1} \\
&= \frac{n^c - \lceil (n^c + 1)(1-\alpha) \rceil + 1}{n^c + 1} \\
&= 1 - \frac{\lceil (n^c + 1)(1-\alpha) \rceil}{n^c + 1},
\end{aligned}
$$

where the second last step is due to the fact that $n^c - \lceil (n^c + 1)(1 - \alpha) \rceil + 1$ integers $u$ satisfy "$u > (n^c + 1)(1 - \alpha)$". Therefore,

$$\mathbb{P}\{Y \in \mathcal{C}(X, \widetilde{Y}; k^c(\alpha)\} = \frac{\lceil (n^c + 1)(1 - \alpha) \rceil}{n^c + 1}.$$

By definition of ceiling and floor functions, it is immediate

$$\frac{(n^c + 1)(1 - \alpha)}{n^c + 1} \leq \frac{\lceil (n^c + 1)(1 - \alpha) \rceil}{n^c + 1} < \frac{(n^c + 1)(1 - \alpha) + 1}{n^c + 1},$$

i.e.,

$$1 - \alpha \leq \frac{\lceil (n^c + 1)(1 - \alpha) \rceil}{n^c + 1} < 1 - \alpha + \frac{1}{n^c + 1},$$

therefore, the conclusion follows.

$\square$

### A.3 PROOF OF PROPOSITION 1 AND THEOREM 4

First, we present two lemmas, which will be used to prove Proposition 1.

**Lemma 2.** *For any sequences $\{a_r\}_{r \in [R]}$ and $\{b_r\}_{r \in [R]}$ of real values,*

$$\prod_{r=1}^{R} a_r - \prod_{r=1}^{R} b_r = \sum_{s=1}^{R} \left( \prod_{r<s} a_r \right) (a_s - b_s) \left( \prod_{r>s} b_r \right).$$

*Proof.* For ease of exposition, define

$$P_{R+1} = \prod_{r=1}^{R} a_r$$

$$P_s = \left( \prod_{r=1}^{s-1} a_r \right) \left( \prod_{r=s}^{R} b_r \right) \quad \text{for} \quad s = 2, \ldots, R;$$

$$P_1 = \prod_{r=1}^{R} b_r.$$

Then

$$\prod_{r=1}^{R} a_r - \prod_{r=1}^{R} b_r$$

$$= (P_{R+1} - P_R) + (P_R - P_{R-1}) + \cdots + (P_3 - P_2) + (P_2 - P_1)$$

$$= \sum_{s=1}^{R} (P_{s+1} - P_s). \tag{A.4}$$

By definition, it is clear that for $s = 2, \ldots, R$,

$$P_s = \left( \prod_{r=1}^{s-1} a_r \right) b_s \left( \prod_{r=s+1}^{R} b_r \right) \text{ and } P_{s+1} = \left( \prod_{r=1}^{s-1} a_r \right) a_s \left( \prod_{r=s+1}^{R} b_r \right),$$

which leads to

$$P_{s+1} - P_s = \left( \prod_{r=1}^{s-1} a_r \right) (a_s - b_s) \left( \prod_{r=s+1}^{R} b_r \right).$$

Then plugging this identity into (A.4) proves the result. $\square$

**Lemma 3.** *Suppose Assumptions 3 and 5 hold. Let* $\mathrm{TV}(P_1, P_2)$ *denote the total variation distance between distributions* $P_1$ *and* $P_2$. *Then for any* $\mathrm{x}_0, \mathrm{x} \in \mathcal{X}$,

$$\mathrm{TV}(P_{\mathrm{x}}, P_{\mathrm{x}_0}) \leq \frac{1}{2}(1 + R)K^{R+1}L\|\mathrm{x} - \mathrm{x}_0\|.$$

*Proof.* By Assumption 5, for all $\mathrm{y} \in [K]$ and all $r \in [R], j \in [K]$,

$$|p_{\mathrm{x}}(\mathrm{y}) - p_{\mathrm{x}_0}(\mathrm{y})| \leq L\|\mathrm{x} - \mathrm{x}_0\| \quad \text{and} \quad \left|q_{\mathrm{x}}^{(r)}(j \mid \mathrm{y}) - q_{\mathrm{x}_0}^{(r)}(j \mid \mathrm{y})\right| \leq L\|\mathrm{x} - \mathrm{x}_0\|. \tag{A.5}$$

Then by Assumption 3,

$$P_{\mathrm{x}}(\mathrm{y}, \widetilde{\mathrm{y}}) := \mathbb{P}(Y = \mathrm{y}, \tilde{Y} = \widetilde{\mathrm{y}} \mid X = \mathrm{x}) = p_{\mathrm{x}}(\mathrm{y}) \prod_{r=1}^{R} q_{\mathrm{x}}^{(r)}(\widetilde{\mathrm{y}}^{(r)} \mid \mathrm{y}),$$

and similarly for $P_{\mathrm{x}_0}(\mathrm{y}, \widetilde{\mathrm{y}})$. Therefore, applying the triangle inequality, we obtain

$$\begin{aligned}
&\left|P_{\mathrm{x}}(\mathrm{y}, \widetilde{\mathrm{y}}) - P_{\mathrm{x}_0}(\mathrm{y}, \widetilde{\mathrm{y}})\right| \\
&= \left|p_{\mathrm{x}}(\mathrm{y}) \prod_{r=1}^{R} q_{\mathrm{x}}^{(r)}(\widetilde{\mathrm{y}}^{(r)} \mid \mathrm{y}) - p_{\mathrm{x}_0}(\mathrm{y}) \prod_{r=1}^{R} q_{\mathrm{x}_0}^{(r)}(\widetilde{\mathrm{y}}^{(r)} \mid \mathrm{y})\right| \\
&= \left|p_{\mathrm{x}}(\mathrm{y}) \prod_{r=1}^{R} q_{\mathrm{x}}^{(r)}(\widetilde{\mathrm{y}}^{(r)} \mid \mathrm{y}) - p_{\mathrm{x}_0}(\mathrm{y}) \prod_{r=1}^{R} q_{\mathrm{x}}^{(r)}(\widetilde{\mathrm{y}}^{(r)} \mid \mathrm{y})\right. \\
&\quad \left. + p_{\mathrm{x}_0}(\mathrm{y}) \prod_{r=1}^{R} q_{\mathrm{x}}^{(r)}(\widetilde{\mathrm{y}}^{(r)} \mid \mathrm{y}) - p_{\mathrm{x}_0}(\mathrm{y}) \prod_{r=1}^{R} q_{\mathrm{x}_0}^{(r)}(\widetilde{\mathrm{y}}^{(r)} \mid \mathrm{y})\right| \\
&\leq \left|p_{\mathrm{x}}(\mathrm{y}) - p_{\mathrm{x}_0}(\mathrm{y})\right| \prod_{r=1}^{R} q_{\mathrm{x}}^{(r)}(\widetilde{\mathrm{y}}^{(r)} \mid \mathrm{y}) + p_{\mathrm{x}_0}(\mathrm{y}) \left|\prod_{r=1}^{R} q_{\mathrm{x}}^{(r)}(\widetilde{\mathrm{y}}^{(r)} \mid \mathrm{y}) - \prod_{r=1}^{R} q_{\mathrm{x}_0}^{(r)}(\widetilde{\mathrm{y}}^{(r)} \mid \mathrm{y})\right| \\
&\leq \left|p_{\mathrm{x}}(\mathrm{y}) - p_{\mathrm{x}_0}(\mathrm{y})\right| + \left|\prod_{r=1}^{R} q_{\mathrm{x}}^{(r)}(\widetilde{\mathrm{y}}^{(r)} \mid \mathrm{y}) - \prod_{r=1}^{R} q_{\mathrm{x}_0}^{(r)}(\widetilde{\mathrm{y}}^{(r)} \mid \mathrm{y})\right|, \tag{A.6}
\end{aligned}$$

because each probability factor $q_{\mathrm{x}}^{(r)}(\cdot \mid \mathrm{y})$ lies in $[0, 1]$ and $p_{\mathrm{x}_0}(\mathrm{y}) \leq 1$.

Applying Lemma 2 with $a_r = q_{\mathrm{x}}^{(r)}(\widetilde{\mathrm{y}}^{(r)} \mid \mathrm{y})$, $b_r = q_{\mathrm{x}_0}^{(r)}(\widetilde{\mathrm{y}}^{(r)} \mid \mathrm{y})$, we obtain

$$\left|\prod_{r=1}^{R} q_{\mathrm{x}}^{(r)}(\widetilde{\mathrm{y}}^{(r)} \mid \mathrm{y}) - \prod_{r=1}^{R} q_{\mathrm{x}_0}^{(r)}(\widetilde{\mathrm{y}}^{(r)} \mid \mathrm{y})\right| \leq \sum_{r=1}^{R} |q_{\mathrm{x}}^{(r)}(\widetilde{\mathrm{y}}^{(r)} \mid \mathrm{y}) - q_{\mathrm{x}_0}^{(r)}(\widetilde{\mathrm{y}}^{(r)} \mid \mathrm{y})|.$$

Combining with (A.5) and (A.6) yields

$$\left|P_{\mathrm{x}}(\mathrm{y}, \widetilde{\mathrm{y}}) - P_{\mathrm{x}_0}(\mathrm{y}, \widetilde{\mathrm{y}})\right| \leq L\|\mathrm{x} - \mathrm{x}_0\| + \sum_{r=1}^{R} L\|\mathrm{x} - \mathrm{x}_0\| = (1 + R)L\|\mathrm{x} - \mathrm{x}_0\|. \tag{A.7}$$

Since the total variation distance between $P_{\mathrm{x}}$ and $P_{\mathrm{x}_0}$ is

$$\mathrm{TV}(P_{\mathrm{x}}, P_{\mathrm{x}_0}) = \frac{1}{2} \sum_{\mathrm{y} \in [K]} \sum_{\widetilde{\mathrm{y}} \in [K]^R} \left|P_{\mathrm{x}}(\mathrm{y}, \widetilde{\mathrm{y}}) - P_{\mathrm{x}_0}(\mathrm{y}, \widetilde{\mathrm{y}})\right|,$$

then using (A.7) and the fact that there are $K$ possible labels and $K^R$ possible annotator combinations, we obtain

$$\mathrm{TV}(P_{\mathrm{x}}, P_{\mathrm{x}_0}) \leq \frac{1}{2}(1 + R)K^{R+1}L\|\mathrm{x} - \mathrm{x}_0\|.$$

$\square$

*Proof of Proposition 1.* For $x \in \mathcal{X}$, define

$$g(x) = \mathbb{P}\{Y \in C_\alpha(X, \widetilde{Y}) \mid X = x\}.$$

Then

$$
\begin{aligned}
\mathbb{E}\{g(X)\} &= \int_{x \in \mathcal{X}} g(x) f_X(x) dx \\
&= \int_{x \in \mathcal{X}} \mathbb{P}\{Y \in C_\alpha(X, \widetilde{Y}) \mid X = x\} f_X(x) dx \\
&= \int_{x \in \mathcal{X}} \left\{ \sum_{(y, \widetilde{y}): y \in C_\alpha(x, \widetilde{y})} f(y, \widetilde{y} \mid x) \right\} f_X(x) dx \\
&= \int_{x \in \mathcal{X}} \sum_{(y, \widetilde{y}): y \in C_\alpha(x, \widetilde{y})} f(y, \widetilde{y} \mid x) f_X(x) dx \\
&= \int_{x \in \mathcal{X}} \sum_{(y, \widetilde{y}): y \in C_\alpha(x, \widetilde{y})} f(y, \widetilde{y}, x) dx \\
&= \mathbb{P}\{Y \in C_\alpha(X, \widetilde{Y})\} \\
&\geq 1 - \alpha, \quad\quad\quad\quad\quad\quad\quad\quad\quad\quad\quad (\text{A.8})
\end{aligned}
$$

where the last step comes from Theorem 3.

Let

$$\mathcal{B}_{\alpha,\gamma} := \mathcal{X} \setminus \mathcal{V}_{\alpha,\gamma} = \left\{ x_0 \in \mathcal{X} : g(x_0) < 1 - \alpha - \gamma \right\}.$$

Then writing $\mathcal{X} = \mathcal{B}_{\alpha,\gamma} \cup \mathcal{V}_{\alpha,\gamma}$, we obtain

$$
\begin{aligned}
\mathbb{E}\{1 - g(X)\} &= \mathbb{E}[\{1 - g(X)\}\{\mathbb{1}(X \in \mathcal{B}_{\alpha,\gamma}) + \mathbb{1}(X \in \mathcal{V}_{\alpha,\gamma})\}] \\
&= \mathbb{E}[\{1 - g(X)\}\mathbb{1}(X \in \mathcal{B}_{\alpha,\gamma})] + \mathbb{E}[\{1 - g(X)\}\mathbb{1}(X \in \mathcal{V}_{\alpha,\gamma})] \\
&= \int \{1 - g(x)\}\mathbb{1}(x \in \mathcal{B}_{\alpha,\gamma}) f_X(x) dx + \int \{1 - g(x)\}\mathbb{1}(x \in \mathcal{V}_{\alpha,\gamma}) f_X(x) dx \\
&= \int_{x \in \mathcal{B}_{\alpha,\gamma}} \{1 - g(x)\} f_X(x) dx + \int_{x \in \mathcal{V}_{\alpha,\gamma}} \{1 - g(x)\} f_X(x) dx \\
&\geq (\alpha + \delta) \int_{x \in \mathcal{B}_{\alpha,\gamma}} f_X(x) dx + 0 \\
&= (\alpha + \delta)\mathbb{P}(X \in \mathcal{B}_{\alpha,\gamma}), \quad\quad\quad\quad\quad\quad\quad\quad\quad\quad (\text{A.9})
\end{aligned}
$$

where we used $1 - g(X) > \alpha + \gamma$ on $\mathcal{B}_{\alpha,\gamma}$ and $1 - g(X) \geq 0$ always.

On the other hand, (A.8) implies

$$\mathbb{E}\{1 - g(X)\} = 1 - \mathbb{E}\{g(X)\} \leq \alpha. \quad\quad\quad\quad (\text{A.10})$$

Combining (A.9) and (A.10) gives

$$\alpha \geq \mathbb{E}[1 - g(X)] \geq (\alpha + \delta)\mathbb{P}(X \in \mathcal{B}_{\alpha,\gamma})$$

leading to

$$\mathbb{P}(X \in \mathcal{B}_{\alpha,\gamma}) \leq \frac{\alpha}{\alpha + \gamma}.$$

Therefore

$$\mathbb{P}(X \in \mathcal{V}_{\alpha,\gamma}) = 1 - \mathbb{P}(X \in \mathcal{B}_{\alpha,\gamma}) \geq 1 - \frac{\alpha}{\alpha + \gamma},$$

which proves the result. □

*Proof of Theorem 4.* For $x \in \mathcal{X}$, let

$$g(x) = \mathbb{P}\{Y \in C_\alpha(X, \widetilde{Y}) \mid X = x\}.$$

For any $x \in \mathcal{N}_{\alpha,\gamma}(\rho)$ and any $\epsilon > 0$, there exists $x_0 \in \mathcal{V}_{\alpha,\gamma}$ such that

$$\|x - x_0\| < \rho + \epsilon.$$

Then by Proposition 3,

$$
\begin{aligned}
|g(x) - g(x_0)| &= \left| \mathbb{P}\{Y \in C_\alpha(X, \widetilde{Y}) \mid X = x\} - \mathbb{P}\{Y \in C_\alpha(X, \widetilde{Y}) \mid X = x_0\} \right| \\
&\leq \mathrm{TV}(P_x, P_{x_0}) \\
&\leq \frac{1}{2}(1 + R)K^{R+1}L\|x - x_0\| \\
&< \frac{1}{2}(1 + R)K^{R+1}L(\rho + \epsilon).
\end{aligned}
$$

Therefore,

$$
\begin{aligned}
g(x) &> g(x_0) - \frac{1}{2}(1 + R)K^{R+1}L(\rho + \epsilon) \\
&> 1 - \alpha - \gamma - \frac{1}{2}(1 + R)K^{R+1}L(\rho + \epsilon),
\end{aligned}
\tag{A.11}
$$

where the last step come from the fact that $g(x^*) \geq 1 - \alpha - \gamma$ for any $x^* \in \mathcal{V}_{\alpha,\gamma}$. Letting $\epsilon \downarrow 0$ gives equation 5.

Finally,

$$
\begin{aligned}
&\mathbb{P}\{Y \in C_\alpha(X, \widetilde{Y}) \mid X \in \mathcal{N}_{\alpha,\gamma}(\rho)\} \\
&= \frac{\mathbb{P}\{Y \in C_\alpha(X, \widetilde{Y}), X \in \mathcal{N}_{\alpha,\gamma}(\rho)\}}{\mathbb{P}\{X \in \mathcal{N}_{\alpha,\gamma}(\rho)\}} \\
&= \frac{\int_{x \in \mathcal{N}_{\alpha,\gamma}(\rho)} \sum_{(y,\tilde{y}):y \in C_\alpha(x,\tilde{y})} f(y, \tilde{y}, x) dx}{\int_{x \in \mathcal{N}_{\alpha,\gamma}(\rho)} f_X(x) dx} \\
&= \frac{\int_{x \in \mathcal{N}_{\alpha,\gamma}(\rho)} \{\sum_{(y,\tilde{y}):y \in C_\alpha(x,\tilde{y})} f(y, \tilde{y} \mid x)\} f_X(x) dx}{\int_{x \in \mathcal{N}_{\alpha,\gamma}(\rho)} f_X(x) dx} \\
&= \frac{\int_{x \in \mathcal{N}_{\alpha,\gamma}(\rho)} P\{Y \in C_\alpha(X, \widetilde{Y}) \mid X = x\} f_X(x) dx}{\int_{x \in \mathcal{N}_{\alpha,\gamma}(\rho)} f_X(x) dx} \\
&= \frac{\int_{x \in \mathcal{N}_{\alpha,\gamma}(\rho)} g(x) f_X(x) dx}{\int_{x \in \mathcal{N}_{\alpha,\gamma}(\rho)} f_X(x) dx} \\
&\geq \frac{\int_{x \in \mathcal{N}_{\alpha,\gamma}(\rho)} \left\{1 - \alpha - \gamma - \frac{1}{2}(1 + R)K^{R+1}L(\rho + \epsilon)\right\} f_X(x) dx}{\int_{x \in \mathcal{N}_{\alpha,\gamma}(\rho)} f_X(x) dx} \\
&= 1 - \alpha - \gamma - \frac{1}{2}(1 + R)K^{R+1}L(\rho + \epsilon)
\end{aligned}
$$

where the second last step is due to equation A.11. When $\rho = 0$, $\mathcal{N}_{\alpha,\gamma}(0) = \mathcal{V}_{\alpha,\gamma}$. $\square$

## B  DETAILS ABOUT PSEUDO-ANCHOR POINTS

Here, we show results for pseudo anchor points.

**Theorem 5.** *If $x$ is a $\delta$-pseudo anchor point with $\delta \in [0, 1)$, then for $r \in [R]$ and $k \in [K]$,*

*(i)* $(1 - \delta)q_x^{(r)}(\tilde{y}^{(r)} \mid k) \leq \mathbb{P}(\widetilde{Y}^{(r)} = \tilde{y}^{(r)} \mid X = x) \leq (K - 1)\delta + q_x^{(r)}(\tilde{y}^{(r)} \mid k);$

*(ii)* $\mathbb{P}(\widetilde{Y}^{(r)} = \tilde{y}^{(r)}|X = x) - (K-1)\delta \le q_x^{(r)}(\tilde{y}^{(r)} \mid k) \le \frac{1}{1-\delta}\mathbb{P}(\widetilde{Y}^{(r)} = \tilde{y}^{(r)}|X = x).$

*Proof.* Assume that x is a $\delta$-pseudo anchor point for class $k$. Then expression (A.1) becomes

$$
\begin{aligned}
&\mathbb{P}(\widetilde{Y}^{(r)} = \tilde{y}^{(r)}|X = x) \\
=\ & \sum_{j\in\mathcal{Y}} \left\{ \mathbb{P}(\widetilde{Y}^{(r)} = \tilde{y}^{(r)}|Y = j, X = x)\mathbb{P}(Y = j|X = x) \right\} \\
=\ & \sum_{j\ne k} \left\{ \mathbb{P}(\widetilde{Y}^{(r)} = \tilde{y}^{(r)}|Y = j, \mathbf{x})\mathbb{P}(Y = j|X = x) \right\} \\
& + \mathbb{P}(\widetilde{Y}^{(r)} = \tilde{y}^{(r)}|Y = k, X = x)\mathbb{P}(Y = k|X = x)
\end{aligned}
\tag{B.1}
$$

Noting that all probabilities in the first term of (B.1) are nonnegative, then by definition of the $\delta$-pseudo anchor point for x, we obtain that

$$
\mathbb{P}(\widetilde{Y}^{(r)}|X = x) \ge (1-\delta)\mathbb{P}(\widetilde{Y}^{(r)} = \tilde{y}^{(r)}|Y = k, X = x).
\tag{B.2}
$$

On the other hand, if x is a $\delta$-pseudo anchor point, then

$$
\mathbb{P}(Y = j|X = x) \le \delta \quad \text{for any} \quad j \ne k,
$$

therefore, by that all conditional probabilities in (B.1) are between 0 and 1, we obtain that

$$
\begin{aligned}
&\mathbb{P}(\widetilde{Y}^{(r)} = \tilde{y}^{(r)}|X = x) \\
\le\ & \sum_{j\ne k} \left\{ \mathbb{P}(\widetilde{Y}^{(r)} = \tilde{y}^{(r)}|Y = j, \mathbf{x}) \times \delta \right\} + q_x^{(r)}(\tilde{y}^{(r)} \mid k)\mathbb{P}(Y = k|X = x) \\
\le\ & \delta \sum_{j\ne k} \mathbb{P}(\widetilde{Y}^{(r)} = \tilde{y}^{(r)}|Y = j, \mathbf{x}) + q_x^{(r)}(\tilde{y}^{(r)} \mid k) \\
=\ & (K-1)\delta + q_x^{(r)}(\tilde{y}^{(r)} \mid k).
\end{aligned}
\tag{B.3}
$$

Combining (B.2) and (B.3) gives us

$$
(1-\delta)q_x^{(r)}(\tilde{y}^{(r)} \mid k) \le \mathbb{P}(\widetilde{Y}^{(r)} = \tilde{y}^{(r)}|X = x) \le (K-1)\delta + q_x^{(r)}(\tilde{y}^{(r)} \mid k),
$$

leading to

$$
\mathbb{P}(\widetilde{Y}^{(r)} = \tilde{y}^{(r)}|X = x) - (K-1)\delta \le q_x^{(r)}(\tilde{y}^{(r)} \mid k) \le \frac{1}{1-\delta}\mathbb{P}(\widetilde{Y}^{(r)} = \tilde{y}^{(r)}|X = x),
\tag{B.4}
$$

which proves (i) and (ii).

$\square$

**Remark.** The inequalities in (B.4) have important implications. In the degenerate situation with $\delta = 0$, i.e., x is an anchor point, (B.4) recovers the identity:

$$
\mathbb{P}(\widetilde{Y}^{(r)} = \tilde{y}^{(r)}|Y = k, X = x) = \mathbb{P}(\widetilde{Y}^{(r)} = \tilde{y}^{(r)}|X = x).
$$

When $\delta$ is extremely small such that $(K-1)\delta$ is close to 0 and $\frac{1}{1-\delta}$ is close to 1, we have

$$
\mathbb{P}(\widetilde{Y}^{(r)} = \tilde{y}^{(r)}|Y = k, X = x) \approx \mathbb{P}(\widetilde{Y}^{(r)} = \tilde{y}^{(r)}|X = x),
$$

showing that a pseudo-anchor point can be practically regarded as an anchor point.

## C  DEEP NEURAL NETWORKS

We describe architectures $\psi_A$, $\psi_C$, and $\psi_S$ in detail. Let

$$\mathcal{S}^{K-1} = \left\{ (s_1, \ldots, s_K)^{\mathsf{T}} : s_j \geq 1 \quad \text{for} \quad j \in [K] \quad \text{and} \quad \sum_{j=1}^{K} s_j = 1 \right\}$$

denote the $(K-1)$-dimensional simplex, and let

$$G : \mathbb{R}^K \longrightarrow \mathcal{S}^{K-1}$$

denote a softmax function, given by

$$G(z) = \begin{pmatrix} \frac{\exp(z_1)}{\sum_{j=1}^{K} \exp(z_j)} \\ \frac{\exp(z_2)}{\sum_{j=1}^{K} \exp(z_j)} \\ \vdots \\ \frac{\exp(z_K)}{\sum_{j=1}^{K} \exp(z_j)} \end{pmatrix} \quad \text{for} \quad z = (z_1, \ldots, z_K)^{\mathsf{T}}.$$

For $r \in [R]$, we now describe the conditional probability mass function of $\widetilde{Y}^{(r)}$, given $Y$ and $X$. Specifically, for $k \in [K]$, we specify the vector of the conditional probability mass functions of $\widetilde{Y}^{(r)}$, given $Y = k$ and $X = x$ as:

$$\begin{pmatrix} \mathbb{P}(\widetilde{Y}^{(r)} = 1 | Y = k, X = x) \\ \mathbb{P}(\widetilde{Y}^{(r)} = 2 | Y = k, X = x) \\ \vdots \\ \mathbb{P}(\widetilde{Y}^{(r)} = K | Y = k, X = x) \end{pmatrix} = G \left\{ \begin{pmatrix} \alpha_1^{(r)} \\ \alpha_2^{(r)} \\ \vdots \\ \alpha_K^{(r)} \end{pmatrix} \psi^{\mathrm{A}}(x) + \begin{pmatrix} \beta_1^{(k)} \\ \beta_2^{(k)} \\ \vdots \\ \beta_K^{(k)} \end{pmatrix} \psi^{\mathrm{C}}(x) \right\}$$

where $\alpha^{(r)} \triangleq (\alpha_1^{(r)}, \ldots, \alpha_K^{(r)})^{\mathsf{T}}$ and $\beta^{(k)} \triangleq (\beta_1^{(k)}, \ldots, \beta_K^{(k)})^{\mathsf{T}}$ are weights; and $\psi^{\mathrm{A}}(x)$ and $\psi^{\mathrm{C}}(x)$ are functions facilitating the dependence on the annotator's skills and the class label.

Expressing this elementwisely, we obtain that for $j \in [K]$,

$$\mathbb{P}(\widetilde{Y}^{(r)} = j | Y = k, X = x) = \frac{\exp\{\langle \alpha_j^{(r)}, \psi^{\mathrm{A}}(x) \rangle + \langle \beta_j^{(k)}, \psi^{\mathrm{C}}(x) \rangle\}}{\sum_{l=1}^{K} \exp\{\langle \alpha_l^{(r)}, \psi^{\mathrm{A}}(x) \rangle + \langle \beta_l^{(k)}, \psi^{\mathrm{C}}(x) \rangle\}}. \tag{C.1}$$

Here, the weights $\alpha^{(r)}$ and $\beta^{(k)}$ and the functions $\psi^{\mathrm{A}}(x_i)$ and $\psi^{\mathrm{C}}(x)$ are unknown, which need to be trained using the data $\overline{\mathcal{D}}_{0,k}$ or its subset.

To flexibly reflect possibly different effects of the annotator expertise ($r$) and the ground truth ($k$) in the annotation process, we employ deep neural network (DNN) architectures to describe $\psi^{\mathrm{A}}(x_i)$ and $\psi^{\mathrm{C}}(x_i)$. Specifically, we specify $\psi^{\mathrm{A}}(x_i)$ as a network with an input layer and an output layer that are linked by $H^{\mathrm{A}} - 1$ hidden layers, where the $h$th hidden layer has $L_h^{\mathrm{A}}$ nodes for $h = 1, \ldots, H^{\mathrm{A}} - 1$. Let $L^{\mathrm{A}} = (L_0^{\mathrm{A}}, L_1^{\mathrm{A}}, \ldots, L_{H^{\mathrm{A}}}^{\mathrm{A}})^{\mathsf{T}}$ denote the width vector for the network, with $L_0^{\mathrm{A}} = p$ for the input layer that records measurements of $p$ elements of $x_i$, and $L_{H^{\mathrm{A}}}^{\mathrm{A}} = 1$ for the output layer. The network architecture $\{H^{\mathrm{A}}, L^{\mathrm{A}}\}$ is characterized by a sequence of linear and nonlinear functions, approximating $\psi^{\mathrm{A}}(x)$ by

$$\widehat{\psi}^{\mathrm{A}}(\theta^{\mathrm{A}}; x) \triangleq W_{H^{\mathrm{A}}}^{\mathrm{A}} \sigma_{H^{\mathrm{A}}-1}^{\mathrm{A}} \left[ \cdots \sigma_2^{\mathrm{A}} \left\{ W_2^{\mathrm{A}} \sigma_1^{\mathrm{A}}(W_1^{\mathrm{A}} x + b_1^{\mathrm{A}}) + b_2^{\mathrm{A}} \right\} + b_3^{\mathrm{A}} \cdots \right] + b_{H^{\mathrm{A}}}^{\mathrm{A}}, \tag{C.2}$$

or equivalently,

$$\widehat{\psi}^{\mathrm{A}}(\theta; x) \triangleq g(H^{\mathrm{A}}; x),$$

where the $g$ functions is determined by the recursive equation

$$g(j; x) = W_j^{\mathrm{A}} g(j-1; x) + b_j^{\mathrm{A}} \quad \text{for} \quad j = 2, \ldots, H^{\mathrm{A}};$$

with

$$g(1; x) = \sigma_1^{\mathrm{A}}(W_1^{\mathrm{A}} x + b_1^{\mathrm{A}}).$$

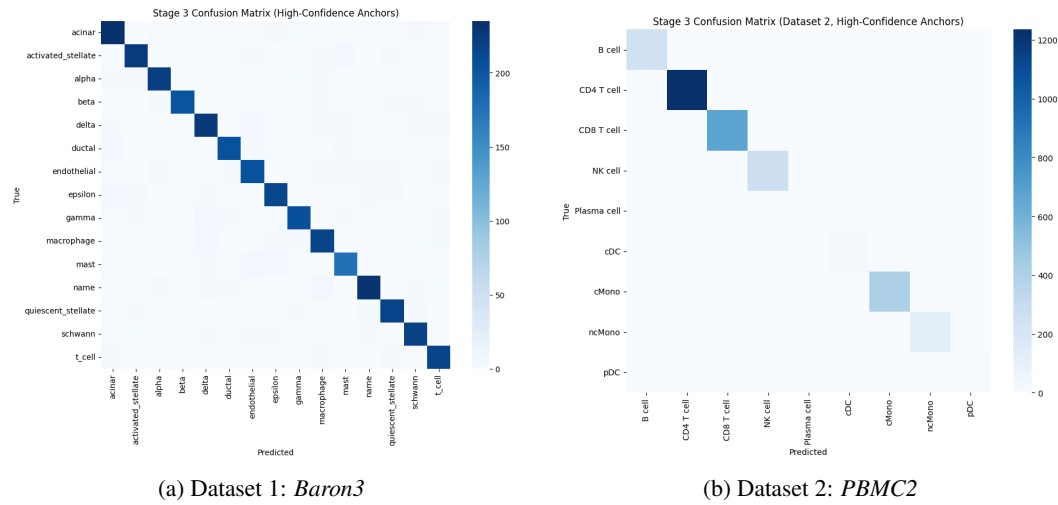

(a) Dataset 1: *Baron3*  (b) Dataset 2: *PBMC2*

Figure 4: Confusion Matrices of predicted cells vs true cell types for both datasets

Here, for $h = 1, \ldots, H^{\mathrm{A}}$, $W_h^{\mathrm{A}}$ is an $L_h^{\mathrm{A}} \times L_{h-1}^{\mathrm{A}}$ weight matrix, $b_h^{\mathrm{A}} \in \mathbb{R}^{L_h^{\mathrm{A}}}$ is the bias vector in layer $h$, $\theta^{\mathrm{A}}$ is the parameter vector formed by stacking $\{W_h^{\mathrm{A}}, b_h^{\mathrm{A}}\}_{h=1}^{H^{\mathrm{A}}}$ from bottom to top, $\sigma_h^{\mathrm{A}}$ is a user-specified activation function that operates elementwise (e.g, a ReLu function), and x is a $p$-dimensional argument.

Analogously, we specify $\psi^{\mathrm{c}}(\mathrm{x})$ as a network, using similar notation but replacing the subscript ᴀ with c for the relevant quantities. Let $\theta = (\theta^{\mathrm{AT}}, \theta^{\mathrm{CT}}; \alpha^{(r)}, \beta^{(k)} : r \in [R], k \in [K])^{\mathrm{T}}$. As a result, the conditional probability in (C.1) is modeled as follows:

$$\mathbb{P}(\widetilde{Y}^{(r)} = j | Y = k, X = x) = \frac{\exp\{\langle \alpha_j^{(r)}, \widehat{\psi}^{\mathrm{A}}(\theta^{\mathrm{A}}; x) \rangle + \langle \beta_j^{(k)}, \widehat{\psi}^{\mathrm{c}}(\theta^{\mathrm{c}}; x) \rangle\}}{\sum_{l=1}^{K} \exp\{\langle \alpha_l^{(r)}, \widehat{\psi}^{\mathrm{A}}(\theta^{\mathrm{A}}; x) \rangle + \langle \beta_l^{(k)}, \widehat{\psi}^{\mathrm{c}}(\theta^{\mathrm{c}}; x) \rangle\}}.$$

# D  EXTENDED RESULTS

We include per–class coverage tables, histograms, and plots to supplement the analysis results in Section 5.

Table 3: Predicted versus annotated cell types

|  | B cell | CD4 T cell | CD8 T cell | NK cell | Plasma cell | cDC | cMono | ncMono | pDC |
|---|---|---|---|---|---|---|---|---|---|
| B cell | 75 | 0 | 0 | 0 | 0 | 0 | 0 | 0 | 0 |
| CD4 T cell | 0 | 363 | 8 | 0 | 0 | 0 | 0 | 0 | 0 |
| CD8 T cell | 0 | 18 | 181 | 4 | 0 | 0 | 0 | 0 | 0 |
| NK cell | 0 | 0 | 5 | 76 | 0 | 0 | 0 | 0 | 0 |
| Plasma cell | 0 | 0 | 0 | 0 | 2 | 0 | 0 | 0 | 0 |
| cDC | 0 | 0 | 0 | 0 | 0 | 6 | 0 | 0 | 0 |
| cMono | 0 | 0 | 0 | 0 | 0 | 0 | 123 | 0 | 0 |
| ncMono | 0 | 0 | 0 | 0 | 0 | 0 | 0 | 36 | 0 |
| pDC | 0 | 0 | 0 | 0 | 0 | 1 | 0 | 0 | 2 |

Table 4: Per–class classification report

|  | B cell | CD4 T cell | CD8 T cell | NK cell | Plasma cell | cDC | cMono | ncMono | pDC | Accuracy | Macro Avg | Weighted Avg |
|---|---|---|---|---|---|---|---|---|---|---|---|---|
| Precision | 1.000 | 0.953 | 0.933 | 0.950 | 1.000 | 0.857 | 1.000 | 1.000 | 1.000 | 0.960 | 0.966 | 0.960 |
| Recall | 1.000 | 0.978 | 0.892 | 0.938 | 1.000 | 1.000 | 1.000 | 1.000 | 0.667 | 0.960 | 0.942 | 0.960 |
| F1-score | 1.000 | 0.965 | 0.912 | 0.944 | 1.000 | 0.923 | 1.000 | 1.000 | 0.800 | 0.960 | 0.949 | 0.960 |
| Support | 75 | 371 | 203 | 81 | 2 | 6 | 123 | 36 | 3 | 0.960 | 900 | 900 |

Table 5: Performance comparison on the *Baron3* dataset using different methods. Metrics include ROC–AUC and PR–AUC.

| Model | ROC–AUC | PR–AUC |
|---|---|---|
| GCN | 0.9942 | 0.9815 |
| MLP | 0.9881 | 0.9799 |
| GAT | 0.9867 | 0.9738 |
| GraphSAGE | 0.9909 | 0.9803 |
| SingleCellNet | 0.9866 | 0.9756 |
| ACTINN | 0.9889 | 0.9804 |
| Proposed Anchor-based CP | 0.9953 | 0.9803 |

Table 6: Comparison of conformal prediction baselines across Baron3 ( 14–cell types) and PBMC2 (Immune dataset). Metrics are empirical coverage and average set size for target coverage levels $\{0.80, 0.85, 0.90, 0.95\}$.

| (a) Baron3 Dataset | | | | | (b) PBMC2 Dataset | | | |
|---|---|---|---|---|---|---|---|---|
| **Method** | **Target** | **Emp. Cov.** | **Set Size** | | **Method** | **Target** | **Emp. Cov.** | **Set Size** |
| APS | 0.80 | 0.967 | 14.49 | | APS | 0.80 | 0.998 | 4.40 |
| | 0.85 | 0.981 | 14.68 | | | 0.85 | 1.000 | 4.94 |
| | 0.90 | 0.986 | 14.74 | | | 0.90 | 1.000 | 5.45 |
| | 0.95 | 0.986 | 14.74 | | | 0.95 | 1.000 | 6.20 |
| RAPS | 0.80 | 0.780 | 10.93 | | RAPS | 0.80 | 0.992 | 4.53 |
| | 0.85 | 0.833 | 11.69 | | | 0.85 | 0.992 | 5.09 |
| | 0.90 | 0.889 | 12.50 | | | 0.90 | 0.992 | 5.40 |
| | 0.95 | 0.947 | 13.32 | | | 0.95 | 0.993 | 5.84 |
| SAPS | 0.80 | 0.815 | 11.39 | | SAPS | 0.80 | 0.903 | 1.00 |
| | 0.85 | 0.860 | 11.93 | | | 0.85 | 0.903 | 1.00 |
| | 0.90 | 0.908 | 12.66 | | | 0.90 | 0.913 | 1.04 |
| | 0.95 | 0.954 | 13.34 | | | 0.95 | 0.958 | 1.36 |
| SoftCP | 0.80 | 0.793 | 10.94 | | SoftCP | 0.80 | 0.927 | 1.00 |
| | 0.85 | 0.860 | 11.87 | | | 0.85 | 0.927 | 1.00 |
| | 0.90 | 0.911 | 12.67 | | | 0.90 | 0.927 | 1.00 |
| | 0.95 | 0.943 | 13.18 | | | 0.95 | 0.953 | 1.09 |
| Top-$k$ | 0.80 | 0.804 | 12.00 | | Top-$k$ | 0.80 | 0.947 | 1.00 |
| | 0.85 | 0.882 | 13.00 | | | 0.85 | 0.947 | 1.00 |
| | 0.90 | 0.949 | 14.00 | | | 0.90 | 0.947 | 1.00 |
| | 0.95 | 0.949 | 14.00 | | | 0.95 | 0.998 | 2.00 |

