# OpenReview forum: "Anchor-Based Conformal Prediction Under Noisy Annotations in Single-Cell Data"
_ICLR.cc/2026/Conference — Submitted to ICLR 2026_

### Official Review · Reviewer_YhNn · 2025-10-14

**Soundness:** 2
**Presentation:** 2
**Contribution:** 2
**Rating:** 2
**Confidence:** 3

**Summary:**

The paper suggests a conformal prediction scheme for the case of a calibration set with multiple annotators.
The method is applied to single-set data.

**Strengths:**

The paper's topic, applying conformal prediction in the presence of label noise, is an important problem that often appears in real-world situations.

**Weaknesses:**

There are many recent studies on conformal prediction with label noise. See e.g. Conformal prediction of classifiers with many classes based on noisy labels, COPA 2025 and the references (from previous years) in that paper.
I find the assumption that we can find anchor points unrealistic in many cases.
The proposed method is not tailored to single-cell data, and it can be validated on other types of datasets.
 The assumptions in Theorem 1  are very strong. What happens if the assumptions are not fulfilled?

**Questions:**

see weeknesses

---

> ### Author Response · Authors · 2025-11-28
>
> **Weaknesses:**
> - **There are many recent studies on conformal prediction with label noise. See e.g. Conformal prediction of classifiers with many classes based on noisy labels, COPA 2025 and the references (from previous years) in that paper. I find the assumption that we can find anchor points unrealistic in many cases. The proposed method is not tailored to single-cell data, and it can be validated on other types of datasets. The assumptions in Theorem 1 are very strong. What happens if the assumptions are not fulfilled?.**
>
> **Our Response:**
> To address your comments on recent studies on conformal prediction with label noise, we have re-written the Introduction section and also included the 2025 reference you mentioned, together with others. While our conformal prediction sets are built based on anchor-point identification, this does not suggest that the associated assumptions are unrealistic for applications. In fact, many existing  approaches impose specific assumptions for the label noise mechanism, such as corrupted by uniform noise with a known noise level (Penso & Goldberger, 2024; Penso et al. 2025), a single one–hot distribution (Stutz et al., 2023), or widely assumed instance–independent label noise. By contrast, we consider a general instance–dependent noise setting that remains relatively under–explored and provides a complementary approach to the literatures.
>
> We suspect that the impression of strong assumptions might arise from Theorems 1 and 2. Those assumptions were in fact importantly overlooked in existing anchor-based methods. One of contributions is to identify and explicitly spell out related assumptions. Such an analysis allows us to fully clarify when those methods are valid and whey they may fail, which provides a theoretical justification of the use of anchor points. Moreover, the assumption in Theorem 1 is not restrictive: it simply requires that given the input, the true label is more likely to be annotated than any other incorrect labels. In other words, annotators are assumed to have reasonably competent skills while they may be unable to annotate the true labels surely. This assumption reflects the fact that noisy data are not always useful automatically; if the data merely contain random noise, they are useless. Learning from noisy data is possible only if the observed labels still contain some informative signal about the underlying truth. To more clearly present our work, we have re-organized the material and added remarks on the practical feasibility of the identified conditions throughout the paper, and we have now explicitly listed associated conditions rather than purely leaving them implicit.
>
> When these assumptions are not fully met, the theoretical guarantees may no longer hold exactly, but the proposed method can still offer an approximate coverage guarantee. This situation is true for any method in the literature: every method has its own assumptions, and when the assumptions fail,   the validity of the method can be compromised.
>
> Finally, in response to  your comments on single-cell data and our method, we have now revised the Introduction section to better highlight how automatic annotation of single-cell data motivates our development of conformal prediction with crowdsourced label noise. While our numerical analyses focus on application to two single-cell datasets to align with the motivating context, our method has a broader application scope to other settings with crowdsourced noisy labels noise, as pointed out in the manuscript.

---

### Official Review · Reviewer_KPgT · 2025-10-30

**Soundness:** 3
**Presentation:** 3
**Contribution:** 3
**Rating:** 4
**Confidence:** 3

**Summary:**

This paper proposes an anchor-based conformal prediction framework for learning predictive models under noisy annotations from multiple annotators, with a focus on single-cell transcriptomics data. The method identifies (pseudo-)anchors—samples with high annotator agreement—to train a base predictor that models annotator-specific noise transitions using deep neural networks. It then calibrates top-$k$ prediction sets to provide distribution-free uncertainty guarantees, ensuring marginal coverage while producing compact sets. Contributions include theoretical guarantees on anchor identification and coverage, as well as empirical validation on two scRNA-seq datasets , demonstrating robustness to label noise.

**Strengths:**

The paper is clear to follow and well-written. The application to single-cell data is of practical importance.

**Weaknesses:**

Dealing with label noise via anchor points and the CP method seems disconnected. The optimization of the model in the first stage does not seem to directly influence CP results.
In addition, calibration based on top-k is not standard and leads to fixed-size sets that does not reflect uncertainty.

**Questions:**

1. Seems that the related work section does not contain a comprehensive discussion on previous CP methods that deal with label noise. Can you add elaborate on that?
2. Can you explain why APS appears to be overly conservative?
3. Can you compare to other scores, such as softmax-based scores, RAPS and SAPS?
4. To strengthen the generality of the proposed method, consider including results on additional commonly used noisy labeled datasets from diverse domains.

---

> ### Author Response · Authors · 2025-11-28
>
> **Weaknesses:**
> - **Dealing with label noise via anchor points and the CP method seems disconnected. The optimization of the model in the first stage does not seem to directly influence CP results. In addition, calibration based on top-k is not standard and leads to fixed-size sets that does not reflect uncertainty.**
>
> **Our Response:**
>  We agree that conformal prediction and anchor-based label-noise modeling address different components of our method, which may appear somewhat disconnected. Often, for a two-stage procedure, one may expect to the impact of the first--stage estimation to be explicitly reflected  in the theoretical results for the second stage. For example, Romano, Sesia, and Candés (2020) commented on this aspect by stating "Our reader will understand that naively substituting $\pi_y(x)$ with $\hat\pi_y(x)$ into the oracle procedure would yield predictions lacking any statistical guarantees because $\hat\pi_y(x)$ may be a poor approximation of $\pi_y(x)$." Here $\pi_y(x)$ represents $\mathbb{P}(Y=y \mid X=x)$, and $\hat\pi_y(x)$ is a suitable estimate of $\pi_y(x)$. However, they also show that such estimation error can be automatically accounted for by adaptively choosing the threshold used to form prediction sets by stating
> "Fortunately, we can automatically account for errors in $\hat\pi_y(x)$ by adaptively choosing the threshold $\tau$ in (3) in such a way as to guarantee finite-sample coverage on future test points".
>
> The same principle also applies to our work, where the variability in the first-stage model training and anchor-point identification is  implicitly reflected  in the formulation of the calibration scores in the second-stage construction of conformal prediction sets. By appropriately choosing a threshold (i.e., $\hat k^c(\alpha)$ in our paper), the finite sample marginal coverage rate is ensured, as detailed in the proof of Theorem 3. In response to your feedback, we have now added comments on this aspect at the end of  Section 2.2.
>
> Regarding the comment on top-$k$ calibration, your perception is not precise. The constructed prediction set (i.e., $C_\alpha(X, \tilde Y)$, or alternatively ${\cal C}(X,\tilde Y; \hat k^c(\alpha))$) does not have a fixed size. Its size is determined  not only by $\hat k^c(\alpha)$ (which is learned from the calibration data) but also by  the actually observed values of $X$ and $\tilde Y$ for the test point. For greater clarity, in this revision we have carefully detailed the implementation procedures in Section 2.2 which appears as follows:
>
> "Further, as the top-$k$ method takes anchor points as input, one might wonder how variability in identifying anchor points, together with uncertainty in model specification and the estimation of model parameters using the likelihood method, may affect statistical guarantees of the resulting prediction sets. This is also a natural concern arising from existing conformal prediction methods, which typically  involves multiple stages of determining intermediate  quantities. Fortunately, as noted by Romanao et al. (2020), these sources of variability are automatically accounted for through the threshold $\hat k^c(\alpha)$, which is chosen adaptively  to ensure finite-sample coverage on future test points, as shown in the proof of Theorem 3 in Appendix A.2.''
>
> **Questions:**
> - **Q1.** **Comment about previous CP methods**
>
> **Our Response:** Thank you for the suggestion, which we take. In preparing the revision, we have re-written the Introduction section and commented on the literature of conformal prediction with label noise.
>
> - **Q2.** **Comment about APS**
>
> **Our Response:**  The  conservativeness of APS could be related to its construction of cumulative-probability–based conformity scores. Because APS includes all labels whose cumulative softmax probabilities exceed a threshold, so when the classifier produces moderate confidence across many classes, APS might systematically over-cover to maintain marginal validity. However, we prefer not to over-interpret the observed patterns, as data heterogeneity plays a pivotal role in affecting the performance of different methods. This perception has also been reflected in our newly added experimental results in Section 5.
>
> - **Q3.** **Comment about other comparisons**
>
> **Our Response:**
> Thank you for the suggestion. In preparing this revision, we have added experimental results obtained from additional baselines: RAPS, SAPS, and SoftCP. Please see details in the last paragraph of Section 5.
>
> - **Q4.** **Comment about generality**
>
> **Our Response:**
> Thank you for this suggestion. In response, we have added remarks on this aspect in Section 6 where we note that it is worthwhile to further assess the performance of the proposed top-k method from other perspectives and other datasets. While the application to two single–cell datasets confirmed the promise of our method, it is useful for assessing how the proposed method performs when applied to other noisy datasets.

---

### Official Review · Reviewer_ub8R · 2025-10-31

**Soundness:** 2
**Presentation:** 3
**Contribution:** 2
**Rating:** 4
**Confidence:** 3

**Summary:**

This paper proposes an anchor-based conformal prediction framework for classification with noisy annotations, motivated by multi-annotator single-cell data. The key idea is to identify anchor samples—instances where multiple annotators strongly agree—then train an annotator-dependent transition model to learn label noise patterns, and finally apply top-k conformal calibration on these anchors to generate uncertainty-calibrated prediction sets. The paper provides theoretical results on anchor identifiability (Theorems 1–2) and standard conformal validity (Theorem 3), and demonstrates empirical results on two single-cell RNA-seq datasets.

**Strengths:**

1. Using multi-annotator agreement to define “anchors” and calibrate predictions is conceptually appealing and relevant for biological or crowdsourced settings.
2. The method achieves coverage close to nominal levels with smaller prediction sets compared to APS baselines.
3. The paper is clear and readable; theoretical statements are mathematically consistent.
4. The anchor-based identifiability concept could inspire broader label-noise research.

**Weaknesses:**

1. The conformal component is entirely standard (split conformal with top-k quantile calibration). There is no novel conformity score, no modified calibration rule, and no new theoretical insight about conformal validity under noisy labels. The main contribution is anchor identifiability, which belongs more to noisy-label learning or multi-annotator modeling rather than to conformal prediction.
2. The paper assumes that coverage guarantees derived on anchor subsets transfer to general noisy test samples, but this is never justified.
3. Exchangeability may not hold once anchors are selected via annotator agreement, so the claimed “distribution-free validity under noisy labels” is overstated.
4. The anchor identifiability theorems are essentially restatements of existing results in label-noise learning (e.g., Xia et al., NeurIPS 2019) under multi-annotator independence assumptions. The conformal theorem is a trivial re-use of standard results from Shafer & Vovk (2008).
5. Experiments are limited to two single-cell RNA-seq datasets; no controlled noise experiments, no cross-domain evaluation (e.g., image or NLP), and no comparison with other noise-aware or conformal baselines. Thus, the empirical evidence is insufficient to support claims of general “noise robustness”.

**Questions:**

1. The conformal component seems standard (split-conformal with top-k quantile calibration). What is the genuine novelty on the conformal prediction side, beyond applying CP to anchors identified from noisy labels?
2. You claim “distribution-free coverage under noisy annotations”. Does this guarantee hold only for the anchor (or pseudo-anchor) subset, or for all test samples? Please clarify the assumed exchangeability after anchor selection.
3. Theorems 1–2 resemble known identifiability results (e.g., Xia et al., NeurIPS 2019). Can you explain what is new in your formulation, and whether the “better-than-random” and independence assumptions can be relaxed or empirically verified?
4. Experiments are limited to two single-cell datasets. Have you tested robustness under controlled synthetic noise or on non-biological datasets to show generality?
5. How does the number or quality of anchors influence coverage and set size?

---

> ### Author Response · Authors · 2025-11-28
>
> - **W1/Q1.** **Comments on the conformal component**\
> **Our Response:** The novelty lies in bridging anchor-based identifiability with split-conformal inference to address prediction uncertainty when the ground-truth class labels are inaccessible.
> While the calibration step follows the classical split-conformal form, our anchor-based construction of conformity scores is new.
> Unlike approaches that rely on specific assumptions for the label noise model, such as
> corrupted by uniform noise with a known noise level (Penso \& Goldberger, 2024), a single one-hot distribution (Stutz et al., 2023), or widely assumed instance-independent label noise, we consider a general case with instance-dependent noise that remains relatively under-explored. Our contributions are complementary to existing conformal prediction methods.
> Furthermore, we identify critical conditions and provide theoretical insights into the use of anchor points. Although anchor points can be heuristically identified based on applying majority–voting to data with noisy labels (e.g., Liu \& Tao, 2016; Patrini et al., 2017), this method is only valid under certain conditions, which, however, have been importantly ignored  in the literature. Our work closes this gap by clearly identifying the settings in which existing methods  utilizing anchor points remain valid.
>
> - **W2.** **Comments on coverage guarantees**\
> **Our Response:**
> We have now carefully re-organized the material for greater clarity. We first present the material about anchor points in Section 2.1,  then describe the proposed method in Section 2.2, and finally provide theoretical guarantees in Theorem 3 (which is now refined).
> We have also theoretically examined the prediction behavior for those features that are not anchor points (in the second half of Section 3).
>
> - **W3/Q2** **Comments on assumed exchangeability**\
> **Our Response:** We have now clarified this aspect in the revised manuscript by explicitly stating Assumption 4 before presenting Theorem 3 which reads reads: "Calibration scores  $\{S(X_i, \widetilde Y_i, Y_i)\}_{i \in {\cal A}^c}$ for the calibration anchor points and "$\{S(X, \widetilde Y, Y)\}$ for a test point are exchangeable and almost surely distinct, or ties are broken at random".
>
> - **W4/Q3.** **Comments on formulation**\
> **Our Response:** Thank you for your comments on these aspects, which,
> however, are not accurate in describing our work. We draw your attention that existing works heuristically used majority-voting to find anchor points from noisy data, but none of them recognized or rigorously justified the validity of this approach. Importantly, this heuristic approach is not automatically valid without requiring conditions. Our work closes this gap by rigorously identifying the associated conditions.
> The proof, just like any work on conformal prediction, looks standard. This is because
> conformal prediction is a model-agnostic framework that applies broadly; its theoretical guarantees under different settings are  derived from common principles and techniques.
>
> - **W5./Q4** **Comments on limited dataset**\
> **Our Response:**
> Our original statement about “noise robustness” aimed to emphasize that our conformal prediction framework is designed to address the practical problem of label noise inherent in cell-type annotation for single-cell RNA-seq data. We appreciate the point that the empirical evidence, currently based on two datasets, does not yet justify broader claims of general noise robustness across domains. In response, we have made revisions to avoid overstating generality and to more clearly situate our contributions within these specific challenges.
> We have updated the Introduction to highlight that single-cell RNA-seq annotation is our motivating application domain, where noisy or imperfect reference labels are common and can significantly hinder downstream analyses. Although the proposed method has the potential to extend to other modalities, our primary goal in this work is to address this domain-specific challenge.
> In Section 5, we also added comparisons with additional conformal and noise-aware baselines, providing a more complete assessment of our method within our context. Finally, in Section 6, we extend the discussion to acknowledge the limitations of the current evaluation and to outline future work.
>
> - **Q5.** **Comments on anchors**\
> **Our Response:**  The following comments are now added in section 6: "As observed in Section 5, different methods may perform differently when applied to different data, and this reflects the fact that heterogeneity in data plays an important role in affecting the performance of a method while the application to  two single-cell datasets confirmed the promise of our method, it is useful in assessing how the proposed method performs when applied to other settings such as imaging or language processing data."

---

### Official Review · Reviewer_yaJG · 2025-11-03

**Soundness:** 2
**Presentation:** 2
**Contribution:** 2
**Rating:** 2
**Confidence:** 4

**Summary:**

This work uses anchor points to develop a conformal prediction framework that produces valid prediction sets in the presence of label noise and multiple noisy annotators. The framework identifies class-specific anchors and models annotator behavior and class dependence using feedforward neural networks. Conformal prediction is used to generate prediction sets with marginal coverage. The paper provides guarantees on existence and identification of anchor points and validates empirical performance on two single-cell RNA-seq datasets.

**Strengths:**

The problem and considered application is interesting and of significance. The experimental setup is interesting, practically relevant, and detailed. The authors perform experiments on single-cell RNA-seq datasets and identify anchors across cell types. The comparison with APS also demonstrates the benefits of the approach.

**Weaknesses:**

1. The paper is not well-written in its current form and needs improvement. The paper mentions concepts without formally introducing them e.g, ‘top-k’ in the abstract and after that as well without explaining the notation; the definition of ‘anchor’ appears late in the paper. Additionally, the introduction is written as related work without appropriately motivating the paper, making the paper not as accessible generally.
2. The paper has hallucinated citations e.g., ‘Anastasios N Angelopoulos et al. Conformal prediction for multi-label classification, ICML 2022’ – I don’t believe there exists any such paper. Additionally, the paper has cited incorrect papers on some instances e.g., p1 l49 ‘split conformal prediction’ – the correct citations for these include Lei et al. (2015); Papadopoulos et al. (2002) as can be seen from Lei et al. (2018). The references are also formatted inconsistently (Anastasios N Angelopoulos et al., 2022; Yaniv Romano et al., 2020 – some references are in this format which is unusual, while others mention all authors). If LLMs were used for generating citations and references, no such usage has been disclosed in the paper and I would like to flag this.
3. Missing/insufficient baselines: The paper compares only with APS which is not expected to perform well in this setup. Not only do there exist score functions which produce smaller sets e.g., RAPS but comparison with methods that are more geared toward similar applications is required to establish benefits of the method.
4. The paper doesn’t study pseudo-anchors or class imbalance and its implications in detail e.g., Plasma cell has 0 anchors (Table 1).

**Questions:**

Missing discussion and comparison with some relevant work:

David Stutz, Abhijit Guha Roy, Tatiana Matejovicova, Patricia Strachan, Ali Taylan Cemgil, Arnaud Doucet. Conformal prediction under ambiguous ground truth. TMLR.

Michele Caprio, David Stutz, Shuo Li, Arnaud Doucet, Conformalized Credal Regions for Classification with Ambiguous Ground Truth, 2024.

---

> ### Author Response · Authors · 2025-11-28
>
> ## **Strengths**
>
> - **The problem and considered application is interesting and of significance. The experimental setup is interesting, practically relevant, and detailed. The authors perform experiments on single-cell RNA-seq datasets and identify anchors across cell types. The comparison with APS also demonstrates the benefits of the approach.**
>
> **Our Response:**
> Thank you for investing your time to review our paper and provide constructive comments and suggestions, which we deeply appreciate. In preparing a revision, we have carefully addressed your comments to improve the presentation of the manuscript.
>
>
> ## **Weaknesses**
>
>  - **W1.** **Comments on presentation.**
>
> **Our Response:**
> To address your comments, we have substantially improved the abstract and Introduction to enhance clarity and accessibility. Please see the updated version for details.
>
> - **W2.** **Comments on references**
>
> **Our Response:**
> We appreciate the reviewer’s careful attention to this citation inconsistency. The errors arose from imperfect coordination among multiple co-authors: some references were initially entered as placeholders, and the inconsistent formats were not fully reconciled before submission. We have now corrected and standardized all citations.
>
> Regarding the reviewer’s question about LLM usage, an earlier submission did include a disclosure that LLMs were used only to help generate placeholder reference entries. This disclosure was unintentionally removed in later revisions by a co-author who was unaware of those placeholders. We apologize for this oversight. In the revised manuscript, we have ensured that all references are accurate and clear.
>
> - **W3.**  **Comments on baselines**
>
> **Our Response:**
> In preparing this revision, we have addressed your comments by adding more experimental results obtained from additional baselines: RAPS, SAPS, and SoftCP. Please see details in the last paragraph of Section 5.
>
> ## **Baron3 Dataset**
>
> | Method | Target | Emp. Cov. | Set Size |
> |--------|--------|-----------|-----------|
> | **APS** | 0.80 | 0.967 | 14.49 |
> |        | 0.85 | 0.981 | 14.68 |
> |        | 0.90 | 0.986 | 14.74 |
> |        | 0.95 | 0.986 | 14.74 |
> | **RAPS** | 0.80 | 0.780 | 10.93 |
> |        | 0.85 | 0.833 | 11.69 |
> |        | 0.90 | 0.889 | 12.50 |
> |        | 0.95 | 0.947 | 13.32 |
> | **SAPS** | 0.80 | 0.815 | 11.39 |
> |        | 0.85 | 0.860 | 11.93 |
> |        | 0.90 | 0.908 | 12.66 |
> |        | 0.95 | 0.954 | 13.34 |
> | **SoftCP** | 0.80 | 0.793 | 10.94 |
> |        | 0.85 | 0.860 | 11.87 |
> |        | 0.90 | 0.911 | 12.67 |
> |        | 0.95 | 0.943 | 13.18 |
> | **Top-k** | 0.80 | 0.804 | 12.00 |
> |        | 0.85 | 0.882 | 13.00 |
> |        | 0.90 | 0.949 | 14.00 |
> |        | 0.95 | 0.949 | 14.00 |
>
> ## **PBMC2 Dataset**
>
> | Method | Target | Emp. Cov. | Set Size |
> |--------|--------|-----------|-----------|
> | **APS** | 0.80 | 0.998 | 4.40 |
> |        | 0.85 | 1.000 | 4.94 |
> |        | 0.90 | 1.000 | 5.45 |
> |        | 0.95 | 1.000 | 6.20 |
> | **RAPS** | 0.80 | 0.992 | 4.53 |
> |        | 0.85 | 0.992 | 5.09 |
> |        | 0.90 | 0.992 | 5.40 |
> |        | 0.95 | 0.993 | 5.84 |
> | **SAPS** | 0.80 | 0.903 | 1.00 |
> |        | 0.85 | 0.903 | 1.00 |
> |        | 0.90 | 0.913 | 1.04 |
> |        | 0.95 | 0.958 | 1.36 |
> | **SoftCP** | 0.80 | 0.927 | 1.00 |
> |        | 0.85 | 0.927 | 1.00 |
> |        | 0.90 | 0.927 | 1.00 |
> |        | 0.95 | 0.953 | 1.09 |
> | **Top-k** | 0.80 | 0.947 | 1.00 |
> |        | 0.85 | 0.947 | 1.00 |
> |        | 0.90 | 0.947 | 1.00 |
> |        | 0.95 | 0.998 | 2.00 |
>
> - **W4.** **Comments on pseudo-anchors and class imbalance.**
>
> **Our Response:**
> Thank you for this comment. We have now included the following discussion in Section 6 to acknowledge this aspect:
> In applications, when the number of anchor points is small or even zero for some classes, we may enlarge $\(\mathcal{A}\)$ by including pseudo-anchors: an instance $\(\mathbf{x}\)$ is called a $\(\delta\)$-pseudo-anchor for class $\(k\)$ if $\(\mathbb{P}(Y=k \mid X=\mathbf{x}) \ge 1-\delta\)$ for $\(0 \le \delta < 1\)$.
> When $\(\delta=0\)$, it becomes an anchor point; $\(\delta\)$-pseudo-anchors are also called anchor points by Xia et al. (2019) if $\(\delta\)$ is close to zero; additional discussions are deferred to Appendix B. A future work is warranted to examine the impact on coverage rates and the sizes of conformal prediction sets when anchor points are mis-identified in settings violating the assumptions in Theorems 1 and 2. Examining the exact influence of the number of anchor points can be valuable, although it is expected that, in principle, the more anchor points, the better the learning results.
>
> ## **Questions**
>
> - **Missing discussion and comparison with some relevant work:**
>
> **Our Response:**
> In the Introduction section of this revision, we have added these references with appropriate discussion.

---

### Author Response · Authors · 2025-11-27

Dear Reviewers:

Thank you all for dedicating your time to review our paper. We appreciated your feedback and suggestions. We are pleased to hear your positive comments, summarized below:

- The problem and considered application is interesting and of significance. The experimental setup is interesting, practically relevant, and detailed.
- Using multi-annotator agreement to define “anchors” and calibrate predictions is conceptually appealing and relevant for biological or crowdsourced settings.
- The method achieves coverage close to nominal levels with smaller prediction sets compared to APS baselines.
- The paper is clear and readable; theoretical statements are mathematically consistent.
- The anchor-based identifiability concept could inspire broader label-noise research.
- The paper is clear to follow and well-written. The application to single-cell data is of practical importance.
- The paper's topic, applying conformal prediction in the presence of label noise, is an important problem that often appears in real-world situations.

Regarding the weakness of our paper, we have carefully addressed each of your comments and suggestions while preparing a revised manuscript; detailed explanations are provided in the responses to each reviewer. Here, we outline the key changes we have made to improve the clarity and presentation of the article:

- The Introduction and abstract have been revised, and new references have been added.
- The main body of the paper has been re-organized to improve the flow of the ideas. In particular, we now begin with a discussion on anchor points and their properties and highlight the differences from existing work in the literature. Although anchor points can be heuristically identified by applying majority voting to data with noisy labels, the conditions for ensuring the validity of this method have been unfortunately ignored in the literature. In this work, we close this gap by identifying the conditions needed to guarantee valid use of anchor points.
- Regarding the theoretical development, we have added new results (Proposition 1 and Theorem 4) and introduced the notion of “conditional valid points” (Definition 3.1). The proofs are provided in Appendix A. The assumptions underlying our development have now been explicitly presented, together with comments on their plausibility and implications.
- New experimental results for additional baselines (RAPS, SAPS, and SoftCP) have been included, with details presented at the end of Section 5.
- Future research directions are outlined and discussed in Section 6.

---

### Meta-Review · Area_Chair_2LXD · 2026-01-06

**Summary:**

- Multiple reviewers questioned the novelty of the conformal prediction component.
- Significant issues were flagged regarding undisclosed LLM usage and citation hallucinations.
- The theoretical assumptions are somewhat strong and potentially unrealistic.
- The empirical evaluation is limited.
- Unclear clarifications or justifications in the presentation.

**Reviewer Concerns:**

In the rebuttal, the authors addressed several technical concerns. They added new theoretical results and included comparisons with additional baselines, expanded the discussion of pseudo-anchors and class imbalance, and improved the organization and clarity of the manuscript. However, fundamental critiques about the limited novelty of the conformal framework, the strong assumptions in the theory, and the narrow empirical validation were not convincingly addressed.

**Reviewer Scores:**

Reviewer yaJG: Likely would have maintained the score due to the ethics violation and citation issues, despite some technical improvements.

Reviewer ub8R: Might have slightly raised the score (e.g., from 4 to 6) given the added baselines and theoretical clarifications, but the core concerns about novelty and limited experiments likely would keep them near the borderline.

Reviewer KPgT: Could have moved from a 4 to a 6 because the authors addressed several specific questions (added baselines, discussion of related work), though the disconnect between stages might still temper enthusiasm.

Reviewer YhNn: Probably would keep the score at 2 (Reject), as the rebuttal did not adequately alleviate concerns about strong assumptions or expanded validation beyond single-cell data.

---

### Decision · Program_Chairs · 2026-01-26

Reject